# Initiation and amplification of SnRK2 activation in abscisic acid signaling

Zhen Lin[1,2,7], Yuan Li[3,7], Yubei Wang[1,2,7], Xiaolei Liu[1], Liang Ma[3], Zhengjing Zhang[1], Chen Mu[1,2], Yan Zhang[1,2], Li Peng[1], Shaojun Xie [4], Chun-Peng Song [5], Huazhong Shi [6], Jian-Kang Zhu [1] & Pengcheng Wang [1,5✉]

The phytohormone abscisic acid (ABA) is crucial for plant responses to environmental challenges. The SNF1-regulated protein kinase 2s (SnRK2s) are key components in ABA-receptor coupled core signaling, and are rapidly phosphorylated and activated by ABA. Recent studies have suggested that Raf-like protein kinases (RAFs) participate in ABA-triggered SnRK2 activation. In vitro kinase assays also suggest the existence of autophosphorylation of SnRK2s. Thus, how SnRK2 kinases are quickly activated during ABA signaling still needs to be clarified. Here, we show that both B2 and B3 RAFs directly phosphorylate SnRK2.6 in the kinase activation loop. This transphosphorylation by RAFs is essential for SnRK2 activation. The activated SnRK2s then intermolecularly trans-phosphorylate other SnRK2s that are not yet activated to amplify the response. High-order *Arabidopsis* mutants lacking multiple B2 and B3 *RAFs* show ABA hyposensitivity. Our findings reveal a unique initiation and amplification mechanism of SnRK2 activation in ABA signaling in higher plants.

[1] Shanghai Center for Plant Stress Biology, CAS Center for Excellence in Molecular Plant Sciences, Chinese Academy of Sciences, Shanghai, China. [2] University of Chinese Academy of Sciences, Beijing, China. [3] State Key Laboratory of Plant Physiology and Biochemistry, College of Biological Sciences, China Agricultural University, Beijing, China. [4] Bioinformatics Core, Purdue University, West Lafayette, IN, USA. [5] Key laboratory of Plant Stress Biology, School of Life Sciences, Henan University, Kaifeng, China. [6] Deperment of Chemistry and biochemistry, Texas Tech University, Lubbock, TX, USA. [7] These authors contributed equally: Zhen Lin, Yuan Li, Yubei Wang. ✉email: pcwang@psc.ac.cn

Environmental challenges like drought, cold, and high salinity induce the accumulation of abscisic acid (ABA), a major stress phytohormone that triggers multiple stress responses in plants[1–6]. ABA controls stomatal closure, seed dormancy and germination, senescence, growth, and development[1–3,7]. The ABA receptor-coupled core signaling pathway has been uncovered[8–10] and consists of three key components: the ABA receptors, the PYRABACTIN RESISTANCE (PYR)/PYR-LIKE (PYL)/REGULATORY COMPONENTS OF ABA RECEPTORS (RCAR) family proteins; the negative regulator clade A type 2 C protein phosphatases (PP2Cs); and the positive regulator SNF1-related protein kinase 2 s (SnRK2s).

Phosphorylation and dephosphorylation determines the activation of SnRK2s and therefore ABA signaling. In the model plant Arabidopsis thaliana, there are ten members of the SnRK2 protein kinase family. Three of them, SnRK2.2, SnRK2.3, and SnRK2.6, are quickly activated within minutes after application of exogenous ABA, while all SnRK2s except SnRK2.9 are activated by osmotic stresses[11–14]. SnRK2.6, also known as OPEN STOMATA 1 (OST1), is mainly expressed in guard cells, while SnRK2.2 and SnRK2.3 are universally expressed[15,16]. The ost1/snrk2.6 mutant shows constitutive stomatal opening and is thus hypersensitive to water deficit[15]. The snrk2.2/2.3/2.6 triple (snrk2-triple) mutant is resistant to ABA and germinates and grows normally at very high concentrations of ABA[13,14]. At least two phosphosites, Ser171 and Ser175, which are located in the activation loop of SnRK2.6, are required for SnRK2.6 activation upon ABA treatment[17,18]. Other phosphosites like the N-terminal Ser7 and Ser29 may also contribute to the activation of SnRK2.6[19,20]. The clade A PP2C phosphatases are the negative regulators of SnRK2s[8–10,21–23]. Under normal growth conditions, PP2Cs inhibit SnRK2.6 by directly binding to and dephosphorylating the Ser175 in the activation loop of SnRK2.6 and block the ABA signaling[20–22].

ABA binds to PYR/PYL/RCARs, and then the ABA-receptor complex inhibits the activity of PP2C phosphatases, resulting in the release of SnRK2s from PP2C-mediated inhibition[9,10,20,24,25]. Using recombinant SnRK2.6 purified from E. coli, Ng et al. (2011) reported that the phosphorylation and activation of SnRK2.6 mainly depends on its autophosphorylation activity[26]. However, whether SnRK2s are auto-activatable in vivo is still debatable since recombinant SnRK2.6 purified from E. coli is already highly phosphorylated and active. Most recently, several studies suggested that dephosphorylated SnRK2.6 and SnRK2.4 have no self-activation activity, and that transphosphorylation of SnRK2s by Raf-like protein kinases (RAFs) is required for SnRK2 activation[27–30]. B2/B3 and B4 RAFs are also called Osmotic stress-activated protein Kinase-100 kDa ($OK^{100}$) and Osmotic stress-activated protein Kinase-130 kDa ($OK^{130}$), respectively, because of their rapid activation by osmotic stress and their molecular weights observed from in-gel kinase assays[27]. The B4 subgroup RAFs ($OK^{130}$) interact with and phosphorylate ABA-independent SnRK2s[27]. In a null mutant of B4 subgroup RAFs, $OK^{130}$-null(raf16/raf40/raf24/raf18/ raf20/raf35/raf42), or raf18/raf20/raf24, the osmotic stress-induced activation of ABA-independent SnRK2s is completely abolished[27,31]. Interestingly, the B3 and B2 RAFs ($OK^{100}$) phosphorylate SnRK2.2, SnRK2.3, and SnRK2.6 in vitro[27,28,30,32]. The high-order mutant OK-quatdec, containing mutations in four B2, three B3, and seven B4 RAFs, shows weak ABA insensitivity in seed germination and root growth[27]. A triple mutant of B3 RAF kinases, m3kδ1/δ6/δ7 (raf5/raf4/raf3), is slightly insensitive to ABA and is impaired in ABA-mediated SnRK2 activation[28]. However, compared to the complete abolishment of ABA responses in snrk2-triple, pyr1pyl1pyl2pyl4-pyl5pyl8 (pyl112458), or pyl-duodecuple (pyr1pyl1pyl2pyl3pyl4-pyl5pyl7pyl8py9pyl10pyl11pyl12, pyl-duodec) mutants[13,14,33,34], the

ABA-insensitivity is much weaker in m3kδ1/δ6/δ7 or OK-quatdec mutants[27,28]. Thus, the role of RAFs in ABA signaling still needs further investigation.

Here, we show that the B2 and B3 subgroup RAFs phosphorylate Ser171 and Ser175 in SnRK2.6 with different specificity and that transphosphorylation is essential for initiating SnRK2.6 phosphorylation and activation. After phosphorylation by RAFs, the activated SnRK2.6 can quickly autophosphorylate (intermolecularly) and activate more SnRK2 proteins. We also generate a series of high-order mutants carrying null mutations in the B2, B3, or both B2 and B3 subgroup RAFs. From phenotypic assays of these high-order mutants, we find that both B2 and B3 subgroup RAFs are essential for ABA signaling. ABA-induced activation of SnRK2.2/2.3/2.6 and ABA-induced gene expression are strongly impaired in $OK^{100}$-oct and $OK^{100}$-nonu mutants lacking 8 and 9 members, respectively, of the B2 and B3 subgroups. $OK^{100}$-oct and $OK^{100}$-nonu also exhibit ABA hyposensitivity and can germinate and grow under extremely high ABA concentrations. We find that ABA does not activate B2 and B3 RAFs; instead, the basal level of RAF kinase activity is essential for SnRK2 activation and initiation of ABA signaling. Our results reveal a crucial RAF-SnRK2 cascade in ABA receptor-coupled core signaling and unique activation machinery for initiating and amplifying stress signaling in higher plants.

## Results

### SnRK2.6 activation requires transphosphorylation by B2 and B3 RAFs in vitro.
Recent studies suggested that several B3 RAFs (RAF3-6) and one B2 RAF (RAF10) can phosphorylate and activate dephosphorylated SnRK2.6 in vitro[27,28,30,32]. In vitro kinase assays and subsequent mass spectrometry revealed that Ser171 and Ser175 of SnRK2.6 might be the major sites for phosphorylation by B2 and B3 subgroup RAFs[27]. To further dissect the role of B2 and B3 RAFs in SnRK2 activation, we first tested the ability and specificity of the recombinant kinase domains (KDs) of B2 and B3 subgroup RAFs in SnRK2.6 phosphorylation. Out of 12 tested B2/B3 RAFs, KDs of 10 RAFs, RAF1-7, and RAF10-12, strongly phosphorylated SnRK2.6$^{KR}$, a kinase-dead form of SnRK2.6 (Fig. 1a). The recombinant KDs of RAF8 and RAF9 had no detectable kinase activity (Supplementary Fig. 1a). The KDs of B2 and B3 RAFs showed distinct specificity for Ser171 and Ser175 of SnRK2.6 in the in vitro assay: the B2 RAFs, RAF7 and RAF10-12, targeted Ser171, while the B3 RAFs, RAF1-6, preferred Ser175 (Fig. 1a).

Ser175 was previously suggested to be a SnRK2.6 autophosphorylation site[26]. Supporting this notion, recombinant SnRK2.6 intermolecularly trans-phosphorylated SnRK2.6$^{KR}$ (Fig. 1b), though the phosphorylation was much weaker than the transphosphorylation by RAF3 and RAF10. Thus, SnRK2.6 can be either transphosphorylated by RAFs or transphosphorylated intermolecularly by other SnRK2.6 molecules.

To further evaluate the role of RAFs in SnRK2.6 activation, we designed an adenosine triphosphate (ATP) analog-based in vitro kinase assay system that distinguishes trans- and autophosphorylation of SnRK2.6 (Fig. 1c). A Met94Gly (M94G) mutation in SnRK2.6 enlarges its ATP binding pocket (Fig. 1c). SnRK2.6$^{M94G}$ can use the ATP analog $N^6$-Benzyl-ATPγS to thiophosphorylate itself or its substrate (Supplementary Fig. 1b). Neither RAF3-KD nor wild-type SnRK2.6 can use the $N^6$-Benzyl-ATPγS as a thiophosphate donor (Fig. 1d, Supplementary Fig. 1b). By this method, thiophosphorylation by activated SnRK2.6$^{M94G}$ can be detected with an anti-thiophosphate ester antibody that only recognizes the thiophosphorylation (Supplementary Fig. 1b). Pre-dephosphorylated SnRK2.6$^{M94G}$ (de-SnRK2.6$^{M94G}$) had no auto-thiophosphorylation activity (Fig. 1d, lanes 2–6, Fig. 1e,

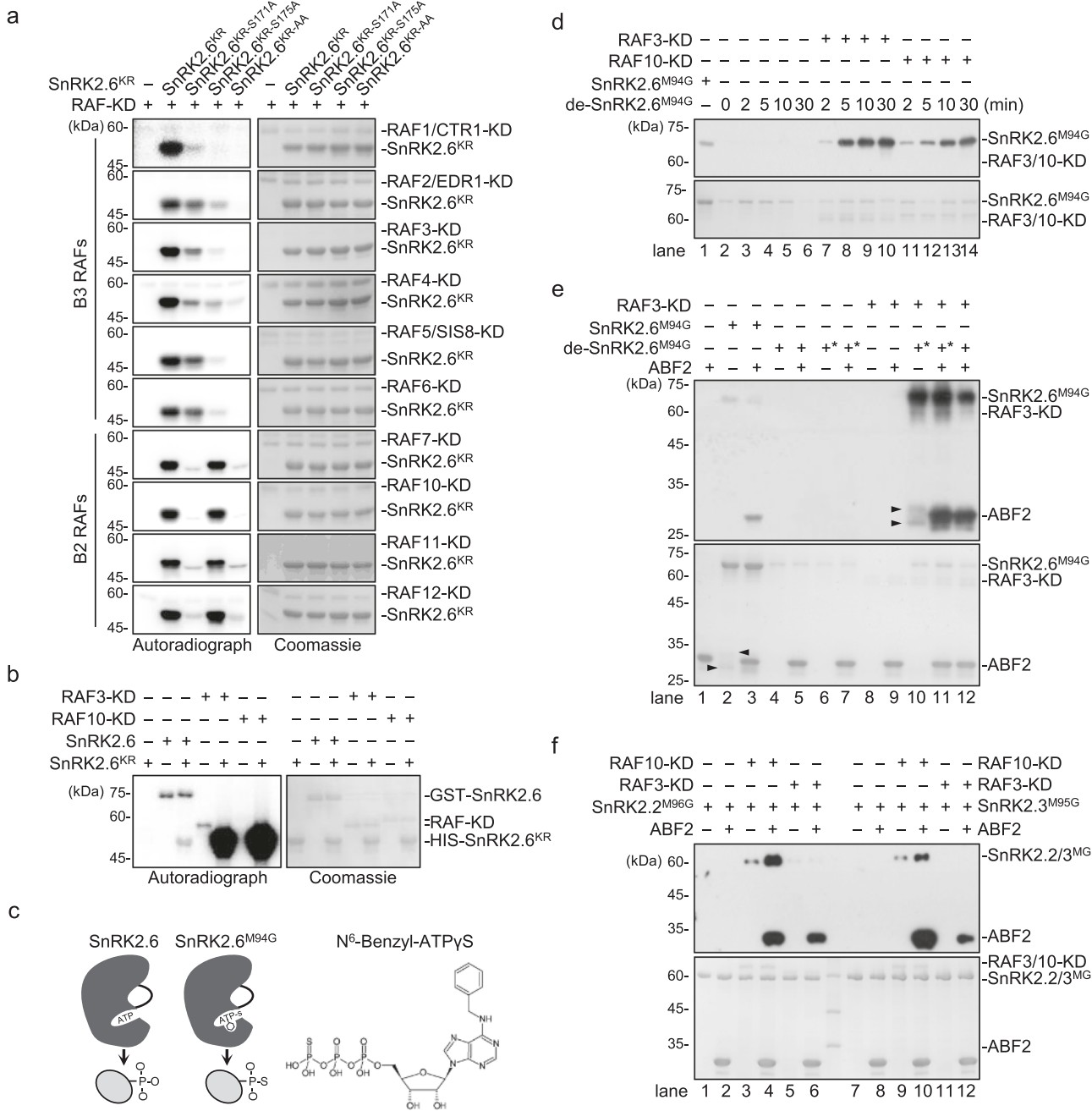

**Fig. 1 SnRK2.6 activation requires transphosphorylation by B2 and B3 RAFs in vitro. a** Recombinant RAF kinase domains (KDs) were used to phosphorylate SnRK2.6$^{KR}$ (SnRK2.6$^{K50R}$, a kinase-dead form of SnRK2.6), SnRK2.6$^{KR}$ with Ser171Ala mutation (SnRK2.6$^{KR-S171A}$), SnRK2.6$^{KR}$ with Ser175Ala mutation (SnRK2.6$^{KR-S175A}$), or SnRK2.6$^{KR}$ proteins with Ser171AlaSer175Ala mutations (SnRK2.6$^{KR-AA}$), in the presence of [γ-$^{32}$p]ATP. Autoradiograph (left) and Coomassie staining (right) show phosphorylation and loading, respectively, of purified GST-RAF-KD and HIS-SnRK2.6$^{KR}$. **b** GST-SnRK2.6, HIS-SUMO-RAF3-KD, and HIS-SUMO-RAF10-KD phosphorylate HIS-SnRK2.6$^{KR}$ in vitro. Recombinant undephosphorylated GST-SnRK2.6 was used to phosphorylate HIS-SnRK2.6$^{KR}$ in the presence of [γ-$^{32}$p]ATP. Autoradiograph (left) and Coomassie staining (right) show phosphorylation and loading, respectively, of purified GST-SnRK2.6, HIS-SUMO-RAF3-KD, HIS-SUMO-RAF10-KD, and HIS-SnRK2.6$^{KR}$. **c** SnRK2.6$^{M94G}$ but not wild type SnRK2.6 can use $N^6$-Benzyl-ATPγS to thiophosphorylate substrate. **d** RAF3-KD and RAF10-KD trigger the autophosphorylation of pre-dephosphorylated GST-SnRK2.6$^{M94G}$ (de-SnRK2.6$^{M94G}$). **e** RAF3-KD activates pre-dephosphorylated GST-SnRK2.6$^{M94G}$ (de-SnRK2.6$^{M94G}$) and the reactivated SnRK2.6$^{M94G}$ phosphorylates itself and ABF2. **f** RAF3-KD and RAF10-KD activate HIS-SUMO-SnRK2.2$^{M96G}$ and HIS-SUMO-SnRK2.3$^{M95G}$. For **d** to **f**, Anti-γ-S immunoblot (upper) and Coomassie staining (lower) show thiophosphorylation and loading, respectively, of recombinant GST-RAF3-KD, GST-RAF10-KD, GST-SnRK2.6$^{M94G}$, HIS-SUMO-SnRK2.2$^{M96G}$, HIS-SUMO-SnRK2.3$^{M95G}$, and GST-ABF2. Asterisks indicate preincubation of SnRK2.6$^{M94G}$ with $N^6$-Benzyl-ATPγS for 30 min before further reaction. Arrows indicate degraded fragments co-purified with GST-SnRK2.6$^{M94G}$. Images shown are representative of at least two independent experiments. Source data are provided in Source Data.

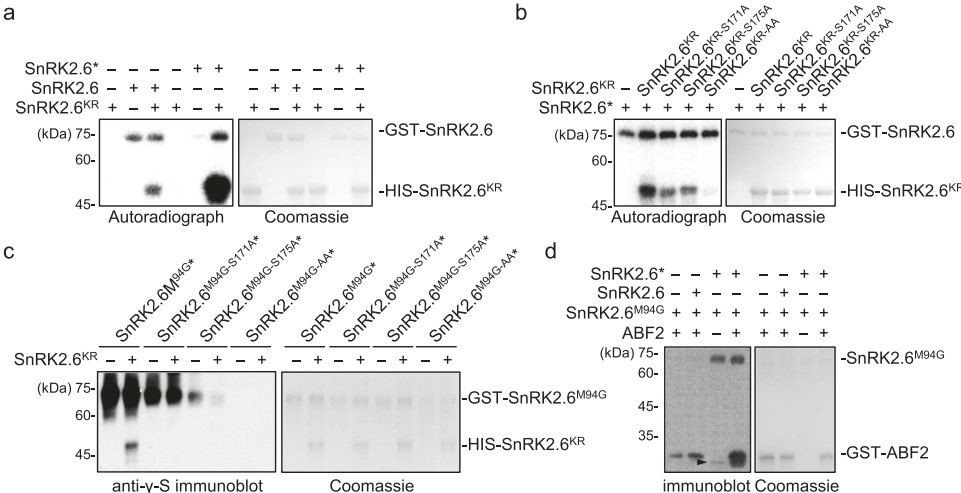

**Fig. 2 Intermolecular transphosphorylation amplifies SnRK2 activation. a** Recombinant SnRK2.6 and pre-activated SnRK2.6 (SnRK2.6*) phosphorylate HIS-SnRK2.6$^{KR}$. Aliquot of SnRK2.6 was pre-incubated with HIS-SUMO-RAF10-KD coated on Ni-NTA beads for 30 min. After removal of HIS-SUMO-RAF10-KD by centrifugation, pre-activated SnRK2.6 (SnRK2.6*) was used to phosphorylate HIS-SnRK2.6$^{KR}$, in the presence of [γ-$^{32}$p]ATP. Same amount of SnRK2.6 without pre-activation was used as control (lanes 1–3). Lane 4 indicates no remaining RAF10-KD after removal. Autoradiograph (left) and Coomassie staining (right) show phosphorylation and loading, respectively, of purified GST-SnRK2.6 and HIS-SnRK2.6$^{KR}$. **b** Pre-activated SnRK2.6 (SnRK2.6*) phosphorylates SnRK2.6$^{KR}$, SnRK2.6$^{KR-S171A}$, SnRK2.6$^{KR-S175A}$, or SnRK2.6$^{KR-AA}$, in the presence of [γ-$^{32}$p]ATP. Autoradiograph (left) and Coomassie staining (right) show phosphorylation and loading, respectively, of purified GST-SnRK2.6 and HIS-SnRK2.6$^{KR}$ proteins. **c** Transphosphorylation activity of pre-activated SnRK2.6$^{M94G}$, SnRK2.6$^{M94G-S171A}$, SnRK2.6$^{M94G-S175A}$, and SnRK2.6$^{M94G-AA}$ on SnRK2.6$^{KR}$. **d** Effect of recombinant SnRK2.6 and pre-activated SnRK2.6 (SnRK2.6*) on SnRK2.6$^{M94G}$ activity. For **c** to **d**, Anti-γ-S immunoblot (left) and Coomassie staining (right) show thiophosphorylation and loading, respectively, of recombinant GST-SnRK2.6$^{M94G}$, HIS -SnRK2.6$^{KR}$, and GST-ABF2. Arrow indicates partial degraded band of GST-SnRK2.6$^{M94G}$. Images shown are representative of at least two independent experiments. Source data are provided in Source Data.

lanes 4–5). Application of recombinant RAF3-KD or RAF10-KD quickly induced the auto-thiophosphorylation of SnRK2.6$^{M94G}$ in a time-dependent manner (Fig. 1d, lanes 7–14), suggesting that transphosphorylation by RAF3 and RAF10 is essential for SnRK2.6$^{M94G}$ auto-thiophosphorylation activity.

We then examined SnRK2.6$^{M94G}$ activity by detecting the thiophosphorylation of ABA-RESPONSIVE ELEMENT-BINDING FACTOR 2 (ABF2), a well-studied SnRK2 substrate. Adding RAF3-KD initiated the thiophosphorylation of both SnRK2.6$^{M94G}$ and ABF2 (Fig. 1e, lanes 10–12). Preincubation with RAF3 significantly enhanced the kinase activity of recombinant SnRK2.6$^{M94G}$ (Fig. 1e, lanes 10–12 compared to lanes 2–3). We also measured the activation effect of RAF3 and RAF10 on SnRK2.2$^{M96G}$ and SnRK2.3$^{M95G}$, the mutated forms that can use the $N^6$-Benzyl-ATPγS. Similar to previous study[26], the recombinant SnRK2.2$^{M96G}$ and SnRK2.3$^{M95G}$ only had weak kinase activity that was rarely detectable in the thiophosphorylation assay (Fig. 1f, lanes 1, 2, 7, and 8). However, adding either RAF10-KD or RAF3-KD strongly enhanced the kinase activities of SnRK2.2$^{M96G}$ and SnRK2.3$^{M95G}$, in the context of ABF2 and SnRK2 thiophosphorylation (Fig. 1f, lanes 4, 6 compared to lanes 2, and lanes 10, 12 compared to lane 8). Taking these results together, transphosphorylation by RAFs is required for the reactivation of SnRK2.2/2.3/2.6. The enhanced thiophosphorylation of SnRK2.6$^{M94G}$, SnRK2.2$^{M96G}$, and SnRK2.3$^{M95G}$ on themselves suggested that activated SnRK2s can quickly intermolecularly transphosphorylate and activate other SnRK2 molecules in vitro.

**Intermolecular transphosphorylation amplifies SnRK2 activation.** To further validate this amplification process, we measured the phosphorylation of SnRK2.6 on SnRK2.6$^{KR}$ by in vitro kinase assay. GST-SnRK2.6 showed weak phosphorylation of HIS-SnRK2.6$^{KR}$ (Fig. 2a). After preincubating with RAF10-KD and ATP for 30 min, and removing RAF10-KD after preincubation,

the activated GST-SnRK2.6 (SnRK2.6*) then showed an enhanced ability to phosphorylate HIS-SnRK2.6$^{KR}$ (Fig. 2a, lane 6). The Ser171Ala or the Ser175Ala mutation significantly impaired, and Ser171AlaSer175Ala double mutation completely abolished, the HIS-SnRK2.6$^{KR}$ phosphorylation by pre-activated GST-SnRK2.6 (Fig. 2b). This result suggests that SnRK2.6 mainly transphosphorylates SnRK2.6 at Ser171 and Ser175 residues, the same major phosphosites of B2 and B3 RAFs.

We further confirmed the trans- and autophosphorylation of SnRK2.6 by mass spectrometry analysis after in vitro kinase reaction. In such a reaction, γ-[$^{18}$O]-ATP was used as the phosphate donor of RAF3 to trans-phosphorylate SnRK2.6$^{M94G}$, and Benzyl-ATPγS was used as the thiophosphate donor for SnRK2.6$^{M94G}$ autophosphorylation. After 30 min of incubation, the reaction was subjected to mass spectrometry analysis. The result confirmed that Ser171 and Ser175 are both intermolecular auto-phosphosites (thio-phosphosites) and trans-phosphosites ($^{18}$O-phosphosites in Supplementary Data 1). Beside Ser171 and Ser175, six other residues, Ser29, Ser43, Ser71, Thr176, Thr179, Try182, in SnRK2.6$^{M94G}$ were both intermolecular auto-phosphosites and trans-phosphosites (Supplementary Data 1). We then generated non-phosphorylatable mutations (Ser/Thr/Try to Ala) on these phosphosites in SnRK2.6 to elucidate their contributions to SnRK2 intermolecularly trans-phosphorylation activity. Mutating either Ser175 or Ser171 completely abolished the ability of SnRK2.6$^{M94G}$ to thiophosphorylate SnRK2.6$^{KR}$ (Fig. 2c), while Thr179 and Try182 may also contribute to the intermolecularly trans-phosphorylation activity of SnRK2.6 (Supplementary Fig. 2a).

We then further verified the ability of pre-activated SnRK2.6 to activate pre-dephosphorylated SnRK2.6$^{M94G}$. We activated GST-SnRK2.6 by incubating it with RAF3-KD and RAF10-KD. After removing RAF3/10-KD, we used the pre-activated SnRK2.6 to phosphorylate SnRK2.6$^{M94G}$. The kinase activity of SnRK2.6$^{M94G}$ was determined by thiophosphorylation of SnRK2.6$^{M94G}$ and

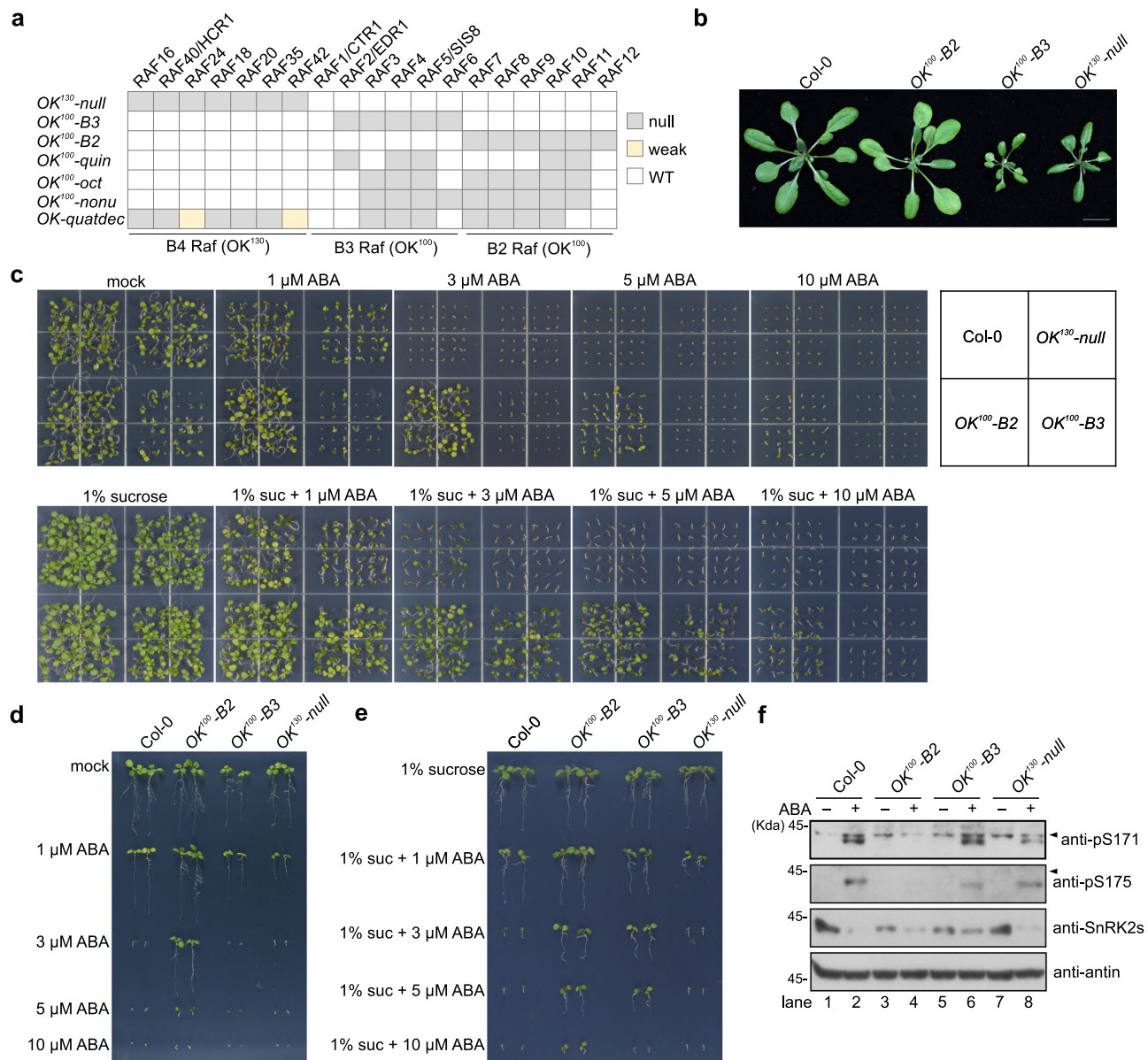

**Fig. 3 B2 and B3 RAFs function redundantly in ABA response in vivo. a** Mutations of RAF genes in the high-order mutants used in this study. **b** Photographs of seedlings after 4 weeks of growth in the soil. Bar = 1 cm. **c** Photographs of seedlings after 10 days of germination and growth on 1/2 MS medium containing indicated concentrations of ABA, without (upper panel) or with (bottom panel) 1% sucrose. The position of mutants in the image is shown in the box at the upper-right corner. **d** Photographs of seedlings after 10 days of germination and growth on 1/2 MS medium containing indicated concentrations of ABA without sucrose. **e** Photographs of seedlings after 10 days of germination and growth on 1/2 MS medium containing indicated concentrations of ABA with 1% sucrose. **f** The ABA-induced phosphorylation of Ser171 and Ser175 of SnRK2.6 in wild-type and the $OK^{100}$-B2 and $OK^{100}$-B3 mutants. The anti-pS171 and anti-pS175 immunoblots were used to show the phosphorylation of the conserved serine residues. Arrows indicate the non-specific bands recognized by anti-pS171 and anti-pS175 antibodies. Images shown are representative of at least two independent experiments. Source data are provided in Source Data.

ABF2. Incubating with pre-activated SnRK2.6 (SnRK2.6*), but not the inactive SnRK2.6, enhanced the kinase activity of SnRK2.6^M94G (Fig. 2d). Pre-activated SnRK2.6 was also able to transphosphorylate SnRK2.2 and SnRK2.3 in the in vitro kinase assay (Supplementary Fig. 2b). Thus, after activation by RAF kinases, SnRK2.6 can intermolecularly autophosphorylate and activate other SnRK2s to amplify the response.

**B2 and B3 RAFs function redundantly in ABA response in vivo.** To validate the RAF-SnRK2 cascade in ABA signaling in planta, we used gene-editing technology to introduce mutations in B2 or B3 subgroup RAF genes in Arabidopsis Col-0 wild type

(Fig. 3a, b). Genotyping thousands of transgenic seeds identified two high-order mutants. $OK^{100}$-B3 [raf2/enhanced disease resistance 1(edr1);raf3;raf4; raf5/sugar insensitive 8 (sis8);raf6] contained null mutations in five of six B3 subgroup RAFs. raf1/ constitutive triple response 1 (ctr1) was not included in the mutant due to a severe morphological phenotype[35]. $OK^{100}$-B2 (raf7;raf8; raf9;raf10;raf11;raf12) contained null mutations in all six B2 subgroup RAF genes (Fig. 3a, b, Supplementary Fig. 3a, b). $OK^{100}$-B3 had a low germination rate on half Murashige and Skoog (MS) medium without sucrose and had strong arrested growth even under normal conditions (Fig. 3c, Supplementary Fig. 4a, b). No ABA insensitivity in germination and seedling

development in $OK^{100}$-B3 was observed on half MS medium without sugar (Fig. 3c, upper panel, d, Supplementary Fig. 4a, b). Interestingly, addition of 1% sucrose strongly improved the germination and seedling development of the $OK^{100}$-B3 mutant, which showed strong ABA insensitivity and geminated on medium containing up to 5 μM ABA (Fig. 3c, e, Supplementary Fig. 4c, d). $OK^{100}$-B2 was hyposensitive to ABA and germinated on medium containing up to 10 μM ABA, with or without sucrose (Fig. 3c–e, Supplementary Fig. 4a–d). The $OK^{130}$-null mutant [raf16;raf40/hydraulic conductivity of root 1(hcr1);raf24; raf18;raf35;raf42] did not show insensitivity to ABA (Fig. 3c–e, Supplementary Fig. 4a–d). $OK^{100}$-B3 and $OK^{100}$-B2 also showed impaired seed dormancy and fresh harvested $OK^{100}$-B3 and $OK^{100}$-B2 seeds had higher germination rates than fresh harvested wild-type seeds (Supplementary Fig. 4e). To measure the phosphorylation and activation of SnRK2s, we used the phosphorylation-specific antibodies recognizing the phosphoserines corresponding to Ser175 and Ser171 in SnRK2.6 (Supplementary Fig. 5 and Supplementary Data 2). ABA-induced phosphorylation of conserved serine residues corresponding to Ser171 and Ser175 in SnRK2.6 was markedly reduced in the $OK^{100}$-B2, and relatively not affected in the $OK^{100}$-B3 and $OK^{130}$-null mutants (Fig. 3f). Thus, B2 and B3 RAFs may have partially redundant roles in ABA responses in plants.

**B2 and B3 RAFs cooperate in ABA-induced SnRK2 activation.** We also generated $OK^{100}$-oct (raf3;raf4;raf5/sis8;raf7;raf8;raf9; raf10;raf11) and $OK^{100}$-nonu (raf3;raf4;raf5/sis8;raf6;raf7;raf8; raf9;raf10;raf11) mutants containing null mutations in eight and nine RAF genes, respectively, of the B2 and B3 subgroups (Fig. 3a, Supplementary Fig. 6). Growth analysis showed that the $OK^{100}$-oct and $OK^{100}$-nonu mutants resembled the previously characterized snrk2-triple mutant and the high-order mutants of PYR/PYL/RCARs exhibiting a strong defect in growth and development (Fig. 4a, Supplementary Fig. 7a–c). By contrast, the $OK^{100}$-quin mutant, which contains mutations in only two B2 and three B3 RAF genes, displayed a wild-type-like growth phenotype (Fig. 4a). The $OK^{100}$-nonu mutant produced very few seeds and had a lower seed germination rate (Supplementary Fig. 7b, c). The $OK^{100}$-oct and $OK^{100}$-nonu mutants showed higher water loss than the wild type (Fig. 4b). Furthermore, ABA-induced stomatal closure was strongly impaired in the $OK^{100}$-oct and $OK^{100}$-nonu mutants, which resembles the snrk2-triple and pyl112458 mutants (Fig. 4c). These results suggested an essential role of the B2 and B3 subgroup RAFs in ABA signaling.

We further evaluated the role of B2 and B3 subgroup RAFs in ABA signaling by assaying seed germination in response to ABA. All tested mutants carrying high-order mutations in B2 and B3 subgroup RAFs, including $OK^{100}$-oct and $OK^{100}$-nonu, showed insensitivity to ABA in seed germination and post-germination seedling growth (Fig. 4d, e, Supplementary Fig. 7d–h). The order of insensitivity to ABA in seed germination was $OK^{100}$-quin < $OK^{100}$-oct = OK-quatdec < $OK^{100}$-nonu < snrk2-triple, pyl112458, and pyl-duodec (Fig. 4d, Supplementary Fig. 7d). Higher-order RAF mutants were clearly more insensitive than lower-order RAF mutants to ABA in seed germination, which further suggests that the RAF members in the B2 and B3 subgroups have essential and partially redundant roles in ABA signaling. $OK^{100}$-oct and $OK^{100}$-nonu also exhibited hyposensitivity to ABA in seedling growth as indicated by root growth, fresh weight measurements, and leaf yellowing (Supplementary Fig. 7g–j).

We then measured ABA-induced SnRK2.2/2.3/2.6 activation in these mutants by in-gel kinase assay. The ABA-induced SnRK2.2/2.3/2.6 activation was almost completely abolished in the $OK^{100}$-oct and $OK^{100}$-nonu mutants, which resembles that in the snrk2-

triple and pyl112458 mutants (Fig. 4f). Interestingly, we observed a weak $OK^{100}$ band (indicated by arrows) in wild type, snrk2-triple and pyl112458 even without ABA or mannitol treatment, and this band was not induced by ABA, when compared to the strong induction by mannitol treatment (Fig. 4f, rightmost lane). Consistently, the immunoblot result showed that the ABA-induced phosphorylation of conserved serine residues corresponding to Ser171 and Ser175 in SnRK2.6 was markedly reduced in the $OK^{100}$-oct and $OK^{100}$-nonu mutants (Fig. 4g). Interestingly, perhaps due to the abolishment of ABA signaling, the ABA-induced rapid SnRK2 degradation was also abolished in the $OK^{100}$-oct and $OK^{100}$-nonu mutants (Fig. 4g). These results strongly indicate that the B2 and B3 subgroup RAFs are essential for ABA-induced SnRK2.2/2.3/2.6 activation.

To evaluate which members of the B2 and B3 subgroups have predominant roles in ABA-regulated seed germination and seedling establishment, we backcrossed $OK^{100}$-nonu with Col-0 wild type and screened F2 populations on 1/2 MS medium containing 10 μM ABA and sucrose. By genotyping 103 individual F2 seedlings with strong ABA insensitivity, we found that each RAF might contribute to ABA hyposensitivity (Chi-square test, $p < 0.05$). RAF3, RAF4, RAF5, and RAF7-9 (closed linked together) might have predominant roles ($p < 0.0001$) in ABA-regulated germination and seedling establishment (Supplementary Table 1).

**B2 and B3 RAFs are required for ABA-induced gene expression.** To investigate the impact of RAF null mutations on ABA-induced gene expression, we performed transcriptomics analysis in WT and $OK^{100}$-oct seedlings. We identified 1368 ABA-induced (>= 3-fold, $p < 0.05$) and 1256 ABA-repressed (>= 3-fold, $p < 0.05$) genes in the wild type. Among these differentially expressed genes (DEGs), only 674 genes were significantly induced (>= 3-fold, $p < 0.05$) and 399 genes were significantly repressed (>= 3-fold, $p < 0.05$) by ABA in the $OK^{100}$-oct seedlings (Fig. 5a, Supplementary Data 3). The heatmap indicated that most ABA-induced and -repressed genes in wild type were less responsive in the $OK^{100}$-oct mutant (Fig. 5b). Quantitative RT-PCR analysis of several ABA-inducible genes, including RESPONSIVE TO ABA 18 (RAB18), COLD-REGULATED 15 A (COR15A), KINASE 1 (KIN1), and RESPONSIVE TO DESICCATION 29B (RD29B), also showed that the induction of these genes by ABA was dramatically impaired in both the $OK^{100}$-oct and $OK^{100}$-nonu mutants (Fig. 5c).

Among the DEGs between wild type and the mutant, 67 genes showed significantly higher expression (>= 3-fold, $p < 0.05$) in the $OK^{100}$-oct mutant under control conditions. Gene Ontology (GO) analysis indicated that these genes are enriched in plant response to chitin, fungus, bacterium, and oxidative stress, suggesting a potential role of the B2 and B3 RAFs in these biotic stress responses (Supplementary Data 4).

To further evaluate the role of B2 and B3 RAFs in ABA-induced gene expression, we used transient activation assays. We used the LUCIFERASE (LUC) reporter gene driven by the ABA-responsive RD29B promoter as an indicator of ABA response[8]. In the wild type, ABA clearly induced the expression of RD29B-LUC in the mesophyll cell protoplasts, while ABA-induced RD29B-LUC expression was completely abolished in the protoplasts of $OK^{100}$-oct, snrk2-triple, pyl112458, and abi1-1 mutants (Fig. 5d). Co-expression of RAF3, RAF5, or RAF11 fully rescued the ABA-induced expression of RD29B-LUC, and co-expression of RAF4, RAF6, RAF7, RAF9, or RAF10 partially rescued the ABA-induced expression of RD29B-LUC in the protoplasts of $OK^{100}$-oct mutant (Fig. 5e). No rescue was seen with co-expression of RAF1, RAF2, RAF8, or RAF12 (Fig. 5e).

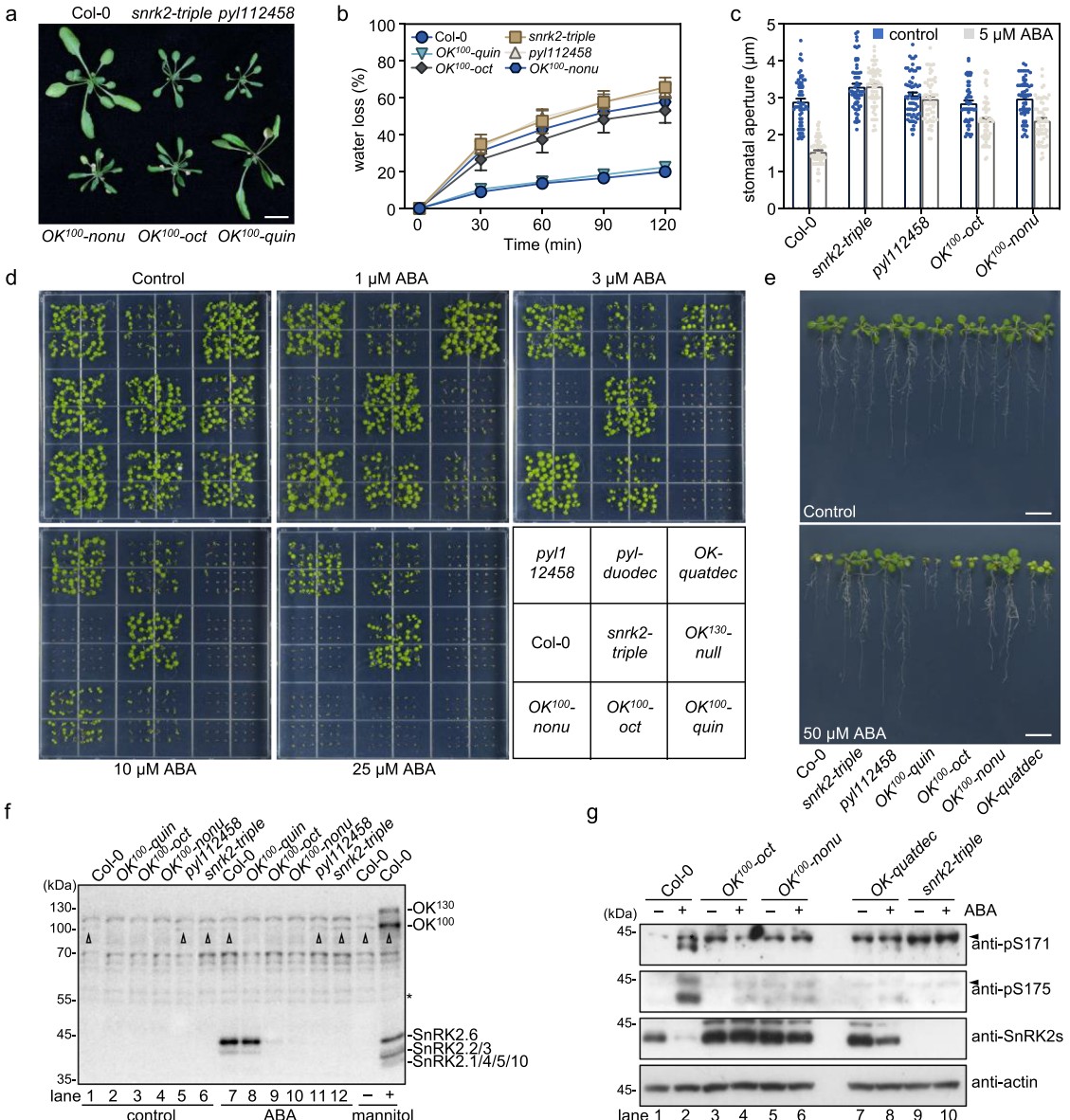

**Fig. 4 B2 and B3 RAFs cooperate in ABA-induced SnRK2 activation. a** Photographs of wild-type and mutant seedlings after 4 weeks of growth in the soil. Bar = 1 cm. **b** Water loss of 4-week-old wild-type and high-order mutants. Error bars, SEM ($n = 5$ biological replicates, each replicate has 5 or 6 individual seedlings). **c** Stomatal closure of 4-week-old wild type and mutants in response to ABA. Error bars, SEM ($n = 59$ or 60 individual stomates). **d** Photographs of seedlings after 10 days germination and growth on 1/2 MS medium containing different concentrations of ABA. The position of mutants in the image is shown in the gray box at the bottom-right corner. **e** Photographs of seedlings growing 10 days after transfer to 1/2 MS medium with or without 50 µM ABA. Bar = 1 cm. **f** In-gel kinase assay showing the activation of SnRK2s and RAFs in wild-type and different mutants with or without 15 min of ABA or mannitol treatment. Arrows indicate the OK[100] band observed in Col-0, *snrk2-triple*, and *pyl112458* mutants. **g** The ABA-induced phosphorylation of Ser171 and Ser175 of SnRK2.6 in wild-type and the OK[100] high-order mutants. Arrows indicate the non-specific bands recognized by anti-pS171 and anti-pS175 antibodies. Images shown are representative of at least two independent experiments. Source data are provided in Source Data.

We then co-transfected SnRK2.2, SnRK2.3, or SnRK2.6 in the transient activation assays with each of RAF1 to RAF12 to evaluate the specificity of different RAFs for different SnRK2 activation (Fig. 5f). Protoplasts co-expressing RAF4, RAF5, RAF6, or RAF9 with SnRK2.2 had higher ABA-induced *RD29B-LUC* expression, while the combination of RAF3, RAF7, RAF8, or RAF10 with SnRK2.6 showed higher ABA-induced *RD29B-LUC* expression in the OK[100]-oct protoplasts. These results suggest that different RAFs exhibit activation specificity for SnRK2.2, SnRK2.3, or SnRK2.6. In addition, RAF3 and RAF7 could not rescue ABA-induced *RD29B-LUC* expression in the *pyl112458* protoplasts (Fig. 5g), which suggests that activation of SnRK2s by RAFs requires ABA-induced release of SnRK2s from PP2C inhibition.

**ABA does not activate B2 and B3 RAFs in plants.** Unlike the strong activation of RAFs by hyperosmolarity, the kinase activity of RAFs was not enhanced by ABA treatment (Fig. 4f). Consistently, the phosphorylation of pSTAGTPEWMAPEVLR, a conserved peptide located in the activation loop of RAF2/EDR1 and RAF3, was not affected by ABA treatment but highly induced by osmotic stress caused by mannitol treatment (Fig. 6a, b, Supplementary Fig. 8a). Multiple phosphosites in this region also could be detected without ABA treatment[27,36] (Fig. 6b, Supplementary Fig. 8a, b, highlighted in Supplementary Data 5). SnRK2.6 showed clearly induced phosphorylation in the peptide containing the phosphorylation site Ser175 by both ABA and mannitol treatments (Supplementary Fig. 8c). We further tested

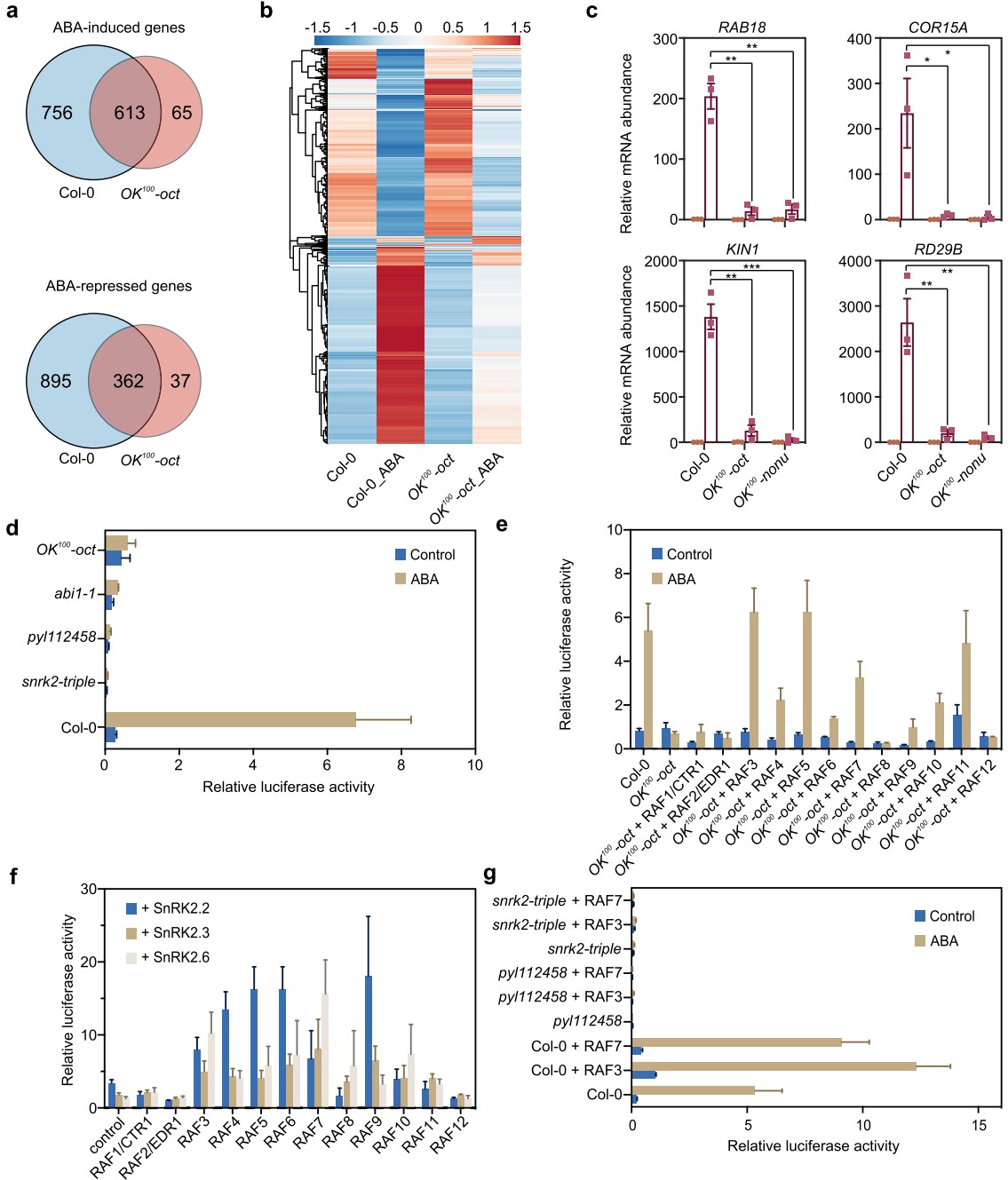

**Fig. 5 B2 and B3 RAFs are required for ABA-induced gene expression. a** Venn diagrams showing the overlaps of ABA-induced and ABA-repressed genes in the wild type and *OK^100-oct* seedlings. **b** Heat map showing the expression levels of ABA-responsive genes in wild type and *OK^100-oct* seedlings. **c** Expression of the ABA-inducible marker genes in wild type, *OK^100-oct*, and *OK^100-nonu* seedlings after 6 h of ABA treatment. Error bars, SEM (*n* = 3 biological replicates). Two-tailed paired *t*-tests, *p < 0.05, **p < 0.01, ***p < 0.001. **d** The activation assay of the ABA-responsive *RD29B-LUC* reporter gene in wild type and mutants. The protoplasts were transformed with the reporter plasmid and incubated with or without 5 μM ABA for 5 h under light. Error bars, SEM (*n* = 3 individual transfections). **e** Add-back assay testing RAF1 to RAF12 in activating the reporter gene in the protoplasts of *OK^100-oct*. Error bars, SEM (*n* = 4 individual transfections). **f** Activation of the reporter gene by the combinations of RAFs with SnRK2.2, SnRK2.3, or SnRK2.6 in the protoplasts of *OK^100-oct*. The ratio of *RD29B-LUC* expression in the protoplasts with 5 μM ABA relative to that without ABA treatment was used to indicate the activation activity of RAF-SnRK2 pairs. Error bars, SEM (*n* = 4 individual transfections). **g** Activation of the reporter gene by RAF3 or RAF7 in the protoplasts of wild type, *pyl112458*, or *snrk2-triple*. Error bars, SEM (*n* = 3 individual transfections). Source data are provided as Source Data files.

whether RAF3 phosphorylation is required for its activity on SnRK2.6 by generating a non-phosphorylatable mutation of RAF3. Co-transfection of RAF3^S763AS766AT770A, with Ser to Ala substitutions in the activation loop, did not rescue the ABA-induced *RD29B-LUC* expression in the protoplasts of *OK^100-oct* (Fig. 6c). RAF3^S763AS766AT770A completely lost its ability to

phosphorylate SnRK2.6 in vitro (Fig. 6d, left panel). However, mutating these conserved residues of RAF10, producing RAF10^T706AT709AT713A, hardly affected RAF10 activity in in vitro kinase or transient expression assays (Fig. 6d, right panel, Supplementary Fig. 8d). Therefore, the activation mechanism of RAF10 in the B2 RAF subgroup may differ from that of RAF3 in

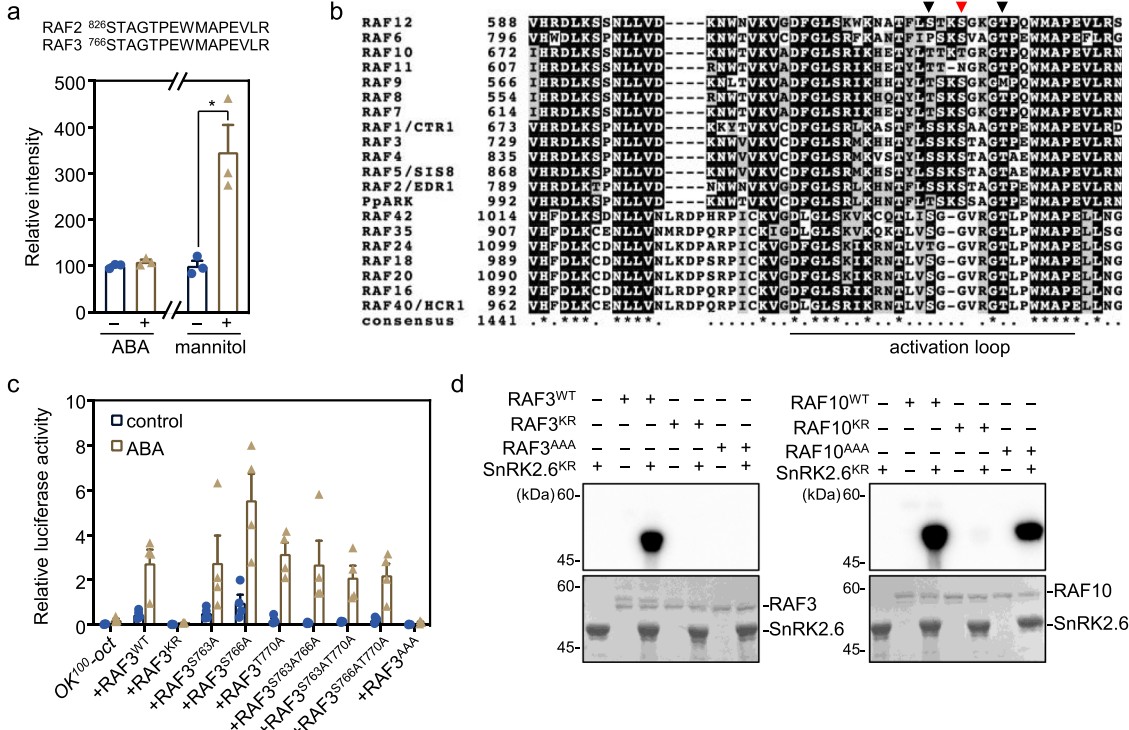

**Fig. 6 ABA does not activate B2 and B3 RAFs in plants. a** The phosphorylation of the conserved phosphosite in RAF2 and RAF3 showing enhanced phosphorylation by mannitol but not ABA treatment. The relative intensity of the phosphopeptide was obtained from previous phosphoproteomics results ($n = 3$ biological replicates). **b** Sequence alignment showing the conserved phosphosites (indicated by arrows) in the activation loop of Arabidopsis B2, B3, B4 RAFs, and PpARK/PpCTR1 from *Physcomitrella patens*. The conserved serine residues corresponding to Ser1029 in PpARK/PpCTR1 are highlighted by the red arrow. **c** Activation of the reporter gene by wild type and the non-phosphorylatable mutants of Ser763, Ser766, and Thr770 in RAF3 in transient reporter gene expression in protoplasts of $OK^{100}$-oct. RAF3$^{K636R}$ (RAF3$^{KR}$), a kinase-dead form of RAF3, is used as a control. Error bars, SEM ($n = 3$ individual transfections). **d** Phosphorylation of SnRK2.6$^{KR}$ by recombinant kinase domain of RAF3 and RAF10. Wild type and mutated recombinant GST-RAF3-KD (left panel) and GST-RAF10-KD (right panel) was used to determine the phosphorylation of SnRK2.6$^{KR}$ expressed and purified from *E. coli* in the presence of [γ-$^{32}$P]ATP. GST-RAF3$^{K636R}$ (RAF3$^{KR}$) and GST-RAF10$^{K515R}$ (RAF10$^{KR}$) were used as negative controls. Autoradiograph (upper) and Coomassie staining (lower) show phosphorylation and loading, respectively, of purified GST-RAF3-KD, GST-RAF10-KD, and HIS-SnRK2.6$^{KR}$. Images shown are representative of at least two independent experiments. Source data are provided in Source Data.

the B3 RAF subgroup. Taken together, these results indicate that B2 and B3 RAFs have basal levels of phosphorylation and activity under normal conditions and that application of ABA does not increase their phosphorylation.

## Discussion

Subgroup B RAFs belong to the MITOGEN-ACTIVATED PROTEIN KINASE KINASE KINASE (MAPKKK) family due to their similarity with animal B-Raf protein kinases[37–39]. In a canonical MAPK cascade, MAPKKKs are activated by extracellular signals and phosphorylate and activate MAPK KINASEs (MAPKKs), which then phosphorylate and activate MAPKs to regulate various cellular processes. Instead of phosphorylating MAPKKs, after rapid activation by osmotic stresses, B2, B3, and B4 RAFs phosphorylate and activate SnRK2s[27–31]. Plants use this noncanonical RAF-SnRK2 cascade to relay early osmotic stress signaling. Here we show a unique initiation-amplification mechanism of RAF-SnRK2 in ABA signaling to ensure rapid activation of SnRK2s with a basal level of Raf kinase activity (Fig. 7). Unlike the B4 subgroup RAFs, which are quickly activated by hyperosmolality[27], the application of exogenous ABA does not enhance the phosphorylation or activity of B2 and B3 subgroup RAFs, as indicated by in-gel kinase assays and phosphoproteomics (Figs. 4 and 6). The phosphoproteomics and the in-gel kinase assay result also revealed the existence of basal-level phosphorylation and activity of B2 and B3 RAFs even

without ABA (Figs. 4f and 6a, Supplementary Data 5). This basal level activation of RAFs might be necessary and sufficient for maintaining the SnRK2 activity and ABA signaling required for normal growth and development[6]. The upstream kinases or other mechanisms for this basal activity of RAFs need to be determined in the future. In the presence of ABA, the ABA and PYR1/PYL/RCAR complex releases SnRK2s from PP2C-mediated inhibition, resulting in the accumulation of uninhibited forms of SnRK2s. The RAFs quickly trans-phosphorylate uninhibited SnRK2s to initiate SnRK2 activation. The activated SnRK2s then intermolecularly transphosphorylate and activate other SnRK2 molecules not yet activated by RAFs, to amplify the ABA signaling (Fig. 7). By this activation-amplification mechanism, the basal level activity of RAF kinases is sufficient to quickly activate SnRK2s to transduce the ABA signal. It would be interesting to apply the ATP analog-based method to evaluate whether this activation-amplification mechanism also exists in other kinase cascades in plants or animal cells, as autophosphorylation is a general feature of many protein kinases.

Several B3 subgroup Raf-like kinases, M3Kδ6/SIS8/RAF5, M3Kδ7/RAF4, M3Kδ1/RAF3, and RAF6, phosphorylate SnRK2.6, and are essential for ABA-induced SnRK2 activation[27,28]. In this study, we show that the B2 subgroup RAFs, together with B3 subgroup, have an essential role in the ABA core signaling pathway. $OK^{100}$-*nonu* and $OK^{100}$-*oct* show strong ABA-insensitivity in germination, leaf yellowing, and stomatal closure. The $OK^{100}$-*nonu* seeds even germinate at an ABA concentration of

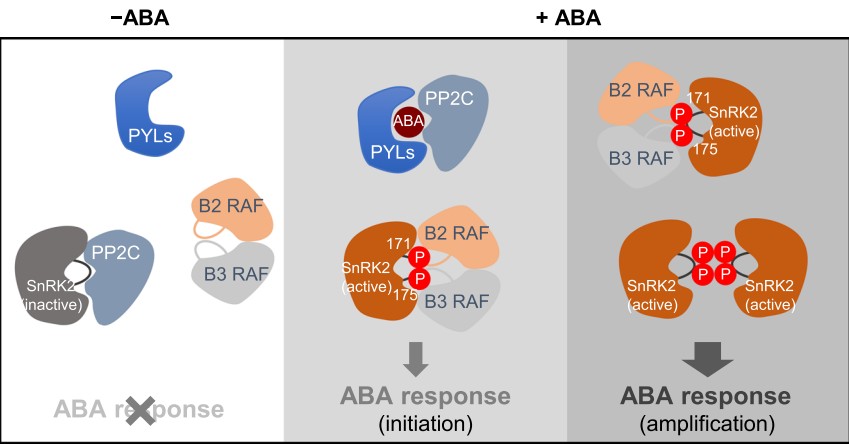

**Fig. 7 A model illustrating the role of B2 and B3 RAFs in ABA signaling in Arabidopsis.** Under unstressed conditions, PP2C binds to and inhibits SnRK2 protein to prevent the transphosphorylation by activated RAFs (left panel). In the presence of ABA, ABA receptor PYR/PYL/RCARs (PYL) complex binds to and inhibits PP2C and SnRK2 is released from PP2C-mediated inhibition. SnRK2 can then be quickly activated by RAFs. In the meantime, stress also activates RAFs by an unknown mechanism (middle panel). The activated SnRK2 can quickly trans-phosphorylate more SnRK2 proteins to amplify the activation and phosphorylate downstream substrates to mediate stress responses (right panel).

up to 25 μM (Fig. 4). To our knowledge, $OK^{100}$-oct and $OK^{100}$-nonu are among the few mutants, including snrk2-triple, pyl112458 and pyl-duodec, that can germinate on such an extremely high concentration of ABA, further supporting the critical role of the B2 and B3 RAFs in ABA responses. However, ABA-insensitivity of $OK^{100}$-nonu is still not identical with that of snrk2-triple and pyl high-order mutants. This suggests that, besides B2 and B3 RAFs, additional protein kinases also participate in ABA-induced SnRK2 activation. At least two protein kinases, BRASSINOSTEROID-INSENSITIVE2 (BIN2) and BRASSINOS-TEROID INSENSITIVE 1-ASSOCIATED RECEPTOR KINASE 1 (BAK1), can phosphorylate SnRK2s and are involved in ABA signaling[40,41]. BIN2 likely phosphorylates only SnRK2.2 and SnRK2.3, but not SnRK2.6[41], at the conserved threonine corresponding to Thr179 in SnRK2.6 (Supplementary Fig. 2). Thus, whether BIN2, or other members of the GSK family, cooperate with RAFs in amplifying ABA-triggered SnRK2 activation, needs to be further studied.

Although both B2 and B3 RAFs are essential for ABA-induced SnRK2 activation, they might have distinct roles in some ABA-regulated biological processes. $OK^{100}$-B3, but not $OK^{100}$-B2, has arrested growth in soil, suggesting a unique role of B3 subgroup RAFs in growth regulation (Fig. 3). $OK^{100}$-B2 shows similar ABA-insensitivity on medium with or without sucrose, whereas $OK^{100}$-B3 only shows strong ABA-insensitivity with exogenous sucrose (Fig. 3). Supporting this notion, the miRNA-M3K was screened from the medium containing sucrose[28,42]. RAF3-5 and RAF7-9 might have more dominant roles in germination and seedling establishment, while RAF3, RAF5, RAF7, and RAF11 rescue the ABA-induced RD29B-LUC expression more robustly. Several key regulators in ABA signaling and synthesis are also involved in sugar responses[43,44]. Mutants raf1/ctr1/sis1, and raf5/sis8, are resistant to high concentrations of sugar[45,46]. Together with these findings, our results suggest a crucial role of B3 subgroup RAFs in sucrose signaling and/or in seed development (e.g., in the accumulation of energy reserve in the seeds). It is notable that the role of RAFs in rescuing ABA-induced RD29B-LUC transcription is not identical to their contribution in gemination and seedling establishment, further indicating that different RAFs have various contributions to different ABA-mediated biological processes. Such functional diversity is also observed in the 14 PYR1/PYL/RCAR ABA receptors. Only pyl112458, but not 3791112 (pyl3/7/9/11/12) shows arrested growth under normal condition[33]. By

contrast, pyl112458 has more predominant roles in ABA-mediated regulation of germination, stomatal movement, etc[33]. In guard cell, PYL2 is sufficient for guard cell ABA-induced responses, whereas in the responses to $CO_2$, PYL4 and PYL5 are essential[47]. PYL8 directly binds to the transcription factor MYB77 to regulate auxin responsive gene expression[48]. PYR1 especially participates in cross-talk between salicylic acid and ethylene, thereby redirecting defense disease resistance towards fungal Plectosphaerella cucumerina[49]. In Arabidopsis, 14 PYLs, at least 8 PP2Cs, three SnRK2s, and 12 members of the B2 and B3 RAF subgroups comprise a complex network in ABA sensing and signaling, which may ensure that plants precisely respond to ever-changing environments. The engineering of ABA receptors is an efficient way to improve stress resistance in both Arabidopsis and crops[50–52]. Our findings regarding B2 and B3 RAFs in stress signaling provide additional targets (e.g., ectopic expression of stress-inducible or constitutively activated forms of RAFs in guard cells) for engineering crops resistant to harsh environmental conditions.

Besides involvement in sugar and ABA signaling, CTR1/RAF1 is a crucial component in ethylene signaling. We excluded RAF1/CTR1 from the $OK^{100}$ high-order mutants because the ctr1 mutant displays severe growth inhibition under normal conditions[35]. However, although the KD of RAF1/CTR1 strongly phosphorylates SnRK2.6 in vitro, neither the full-length RAF1/CTR1 nor RAF1/CTR1-KD rescued the ABA-induced expression of RD29B-LUC in the protoplasts of $OK^{100}$-oct (Supplementary Fig. 8e). Thus, additional mechanisms may determine RAF1 specificity in vivo. Similarly, RAF2, RAF8, and RAF12 only show weak activities on the induction of RD29B-LUC expression in the protoplasts. The roles of these RAFs in ABA signaling therefore need to be further investigated.

The phosphorylation of Ser1029 of the ABA AND ABIOTIC STRESS-RESPONSIVE RAF-LIKE KINASE (PpARK)/PpCTR1 in Physcomitrella patens is induced by exogenous ABA in P. patens[53–55], which is inconsistent with our observations on RAF3 and RAF10 (Fig. 6). Therefore, P. patens and higher plants may adopt different machinery to relay ABA signaling. In addition, non-phosphorylatable mutations at Ser1029 in PpARK/PpCTR1, or Ser763Ser766AThr770 in RAF3, abolished their kinase activities, suggesting phosphorylation-dependent activation of PpARK/PpCTR1 and RAF3. By contrast, the activation of RAF10 might be independent of phosphorylation. In animal cells, RAF

kinases can be activated through phosphorylation, dimerization, or by binding of small GTPases, scaffold protein, 14-3-3 proteins, etc.[56–58]. Future work will investigate the phosphorylation or other activation mechanisms of RAF-SnRK2 cascades in different plant species and their roles in plant adaptive plasticity.

## Methods

**Seed germination and plant growth assay**. Seeds were surface-sterilized in 70% ethanol for 10 min, followed by four times washing with sterile-deionized water. For the germination assay, seeds were sown on 1/2 Murashige and Skoog (MS) medium (0.75% agar, pH 5.7) with or without the indicated concentrations of ABA and 1% sucrose. Plates were kept at 4 °C for 3 days in darkness for stratification and then shifted to a plant growth chamber set at 23 °C and a 16 h light/8 h dark photoperiod. After 72 h of stratification, radical emergence was examined, and photographs of seedlings were taken at the times indicated. For growth assays, seeds were placed on 1/2 MS medium (0.75% agar, pH 5.7) and plates were placed vertically in a plant growth chamber after 3 days of stratification. After 3–4 days, the seedlings were transferred to medium with or without the indicated concentrations of ABA. Root length and fresh weight were measured at the indicated days. For seed dormancy assays, fresh seeds were harvested and sown on 1/2 MS medium (0.75% agar, pH 5.7) and plates were placed in a plant growth chamber. Radical emergence was measured 48 h after transferring.

**Generation of OK$^{100}$ high-order mutants**. The clustered regularly interspaced short palindromic repeats/CRISPR-associated 9 (CRISPR-Cas9) and guide RNA fragment from pCAMBIA-2300-11RAFs[27] was cloned into pCAMBIA-1300. The resulting vectors containing sgRNAs targeting B2, B3, or B2/B3 RAFs were used to transform wild type to generate OK$^{100}$-B2, OK$^{100}$-B3, OK$^{100}$-oct, and OK$^{100}$-nonu. The transgenic plants were screened for hygromycin resistance. The T1 transformants were identified by sequencing the fragments with the RAF target regions, which were amplified by PCR using primer pairs listed in Supplementary Data 6.

**In-gel kinase assay**. For in-gel kinase assays, 20 μg extract of total proteins were electrophoresed on 10% SDS/PAGE embedded with histone as a substrate for kinase. The gel was then washed three times at room temperature for 30 min each with washing buffer (25 mM Tris-Cl, pH 7.5, 0.5 mM Dithiothreitol (DTT), 0.1 mM Na$_3$VO$_4$, 5 mM NaF, 0.5 mg/mL BSA, and 0.1% Triton X-100). The kinase was allowed to renature in renaturing buffer (25 mM Tris-HCl, pH 7.5, 1 mM DTT, 0.1 mM Na$_3$VO$_4$, and 5 mM NaF) and incubated at 4 °C overnight with three changes of renaturing buffer. The gel was further incubated at room temperature in 30 mL reaction buffer (25 mM Tris-Cl, pH 7.5, 2 mM EGTA, 12 mM MgCl$_2$, 1 mM DTT, and 0.1 mM Na$_3$VO$_4$) with 200 nM ATP plus 50 μCi of [γ-$^{32}$P]ATP for 90 min. The reaction was stopped by transferring the gel into 5% (w/v) trichloroacetic acid and 1% (w/v) sodium pyrophosphate. The gel was then washed to remove unincorporated [γ-$^{32}$P]ATP in the same solution for at least 5 h with five changes. Radioactivity was detected with a Personal Molecular Imager (Bio-Rad).

**RNA sequencing and data analysis**. Total RNA was isolated from two-week-old seedlings of Col-0 and OK$^{100}$-oct mutant, with and without ABA treatment, using RNeasy Plant Mini Kit (Qiagen). Total RNA (1 μg) was used for library preparation with NEBNext Ultra II Directional RNA Library Prep Kit for Illumina (New England BioLabs, E7765) following the manufacturer's instructions. Prepared libraries were assessed for fragment size using NGS High-Sensitivity kit on a Fragment Analyzer (AATI), and for quantity using Qubit 2.0 fluorometer (Thermo Fisher Scientific) and KAPA Library Quantification Kit (Kapa, KK4824). All libraries were sequenced in paired-end 150 bases protocol (PE150) on an Illumina Nova sequencer.

The paired-end reads were cleaned by Trimmomatic[59] (version 0.39). After trimming the adapter sequence, removing low quality bases, and filtering short reads, clear read pairs were retained for further analysis. The *Arabidopsis thaliana* reference genome sequence was downloaded from TAIR10. Clean reads were mapped to the genome sequence by HISAT (2.1.0)[60] with default parameters. Number of reads that were mapped to each gene was calculated with the htseq-count script in HTSeq (0.11.2)[61]. EdgeR[62] was used to identify genes that were differentially expressed. Genes with at least three-fold change in expression and with an FDR < 0.05 were considered differentially expressed genes (DEGs).

**Analysis of gene expression by qRT-PCR**. Total RNA was extracted from two-week-old wild-type, OK$^{100}$-oct, and OK$^{100}$-nonu seedlings with or without 50 μM ABA treatment for 6 h. Total RNA was isolated using the RNeasy Plant Mini Kit (Qiagen) according to the manufacturer's instructions. Genomic DNA was removed using RNase-free DNase and subsequently, 1 μg of total RNA was reverse transcribed using the iScript$^{TM}$ gDNA Clear cDNA Synthesis Kit (Bio-Rad) following the manufacturer's instructions. The actin gene was used as an internal control. Quantification was performed using three independent biological replicates.

**Water loss measurement**. The water loss was estimated on detached rosette leaves of 4-week-old plants by weighing using a weighing dish. Leaves were then kept on the laboratory bench for at least 30 min. Fresh weight was monitored before and after the procedure and at the times indicated. Water loss was expressed as a percentage of initial fresh weight.

**Stomatal bioassay**. For stomatal aperture assay, rosette leaves of 4-week-old Arabidopsis seedlings were taken. Epidermal strips were peeled out and incubated in buffer containing 50 mM KCl, 10 mM MES, pH 6.15, in a plant growth chamber for 3 h before ABA treatment. Stomatal apertures were measured 2 h after the addition of 5 μM ABA. The apertures of about 60 stomata per sample were measured by quantifying the pore width of stomata using Image J software (1.51 K). All the experiments were repeated at least three times.

**Protein purification and in vitro kinase assay**. For in vitro kinase assays, full-length coding sequence of SnRK2.6 and kinase domains of RAFs were cloned into either pGEX-4T-1, pET28a or pET-SUMO vectors and transformed into BL21 or ArcticExpression cells. The recombinant proteins were expressed and purified using standard protocols. For the phosphorylation assay, recombinant kinase domains of RAFs (aa541-821 for RAF1-KD, aa650-933 for RAF2-KD, aa600-880 for RAF3-KD, aa710-992 for RAF4-KD, aa733-1030 for RAF5-KD, aa645-956 for RAF6-KD, aa470-773 for RAF7-KD, aa424-671 for RAF8-KD, aa436-730 for RAF9-KD, aa466-767 for RAF10-KD, aa472-765 for RAF11-KD, aa457-735 for RAF12-KD) were incubated with "kinase-dead" forms of SnRK2.6 with or without Ser to Ala mutations at Ser171 and Ser175 in reaction buffer (25 mM Tris HCl, pH 7.4, 12 mM MgCl$_2$, 2 mM DTT), with 1 μM ATP plus 1 μCi of [γ-$^{32}$P] ATP for 30 min at 30 °C. Reactions were stopped by boiling in SDS sample buffer and proteins were separated by 10% SDS-PAGE.

For the dephosphorylation assay, SnRK2.6$^{M94G}$ coated on Glutathione Sepharose (Cytiva) were dephosphorylated with Lambda Protein Phosphatase (λPP, NEB, P0753S) for 30 min and the λPP was removed by washing three times with protein buffer (25 mM Tris HCl, pH 7.4, 150 mM NaCl). To detect the effects of RAF3-KD and RAF10-KD on SnRK2.6$^{M94G}$, SnRK2.2$^{M96G}$, and SnRK2.3$^{M95G}$ thiophosphorylation and activity, recombinant GST-RAF3/10-KD was incubated with pre-dephosphorylated SnRK2.6$^{M94G}$, SnRK2.2$^{M96G}$, or SnRK2.3$^{M95G}$ for 30 min in reaction buffer (25 mM Tris HCl, pH 7.4, 12 mM MgCl$_2$, 2 mM MnCl$_2$, 0.5 mM DTT, 50 μM ATP, 50 μM N$^6$-Benzyl-ATPγS). Then ABF2 was added to the reaction and incubated for an additional 30 min. This phosphorylation reaction was stopped by adding EDTA to a final concentration of 25 μM. A final concentration of 2.5 mM p-nitrobenzyl mesylate (Abcam, ab138910) was added to proceed the alkylating reaction for 1 h at room temperature. Samples with SDS sample buffer were boiled and separated by SDS-PAGE, transferred to Polyvinylidene fluoride (PVDF) membrane, and immunoblotted with antibodies against thiophosphate ester (Abcam, ab92570). To pre-activate SnRK2.6, HIS-SUMO-RAF-KD proteins were coated on the Ni-NTA beads and incubated with SnRK2.6 (in solution) in the presence of ATP. HIS-SUMO-RAF-KD were removed by centrifuging after the reaction.

**Protoplast isolation and transactivation assay**. Protoplasts were isolated from leaves of 4-week-old plants grown under a short photoperiod (10 h light at 23 °C/14 h dark at 20 °C). Leaf strips were excised from the middle parts of young rosette leaves, dipped in enzyme solution containing cellulase R10 (Yakult Pharmaceutical Industry) and macerozyme R10 (Yakult Pharmaceutical Industry) and incubated at room temperature in the dark. The protoplast solution was diluted with an equal volume of W5 solution (2 mM MES, pH 5.7, 154 mM NaCl, 125 mM CaCl$_2$, and 5 mM KCl) and filtered through a nylon mesh. The flow-through was centrifuged at 100 g for 2 min to pellet the protoplasts. Protoplasts were resuspended in W5 solution and incubated for 30 min. 100 μL of protoplasts suspended in MMG solution (4 mM MES, pH 5.7, 0.4 M mannitol, and 15 mM MgCl$_2$) were mixed with the plasmid mix and added to 110 μL PEG solution (40% w/v PEG-4000, 0.2 M mannitol, and 100 mM CaCl$_2$). The transfection mixture was mixed completely by gently tapping the tube followed by incubation at room temperature for 5 min. The protoplasts were washed twice with 1 mL W5 solution. After transfection, protoplasts were left for incubation for a further 5 h under light in washing and incubation solution (0.5 M mannitol, 20 mM KCl, 4 mM MES, pH 5.7) with or without 5 μM ABA. The RD29B-LUC (7 μg of plasmid per transfection) and ZmUBQ-GUS (1 μg per transfection) were used as an ABA-responsive reporter gene and as an internal control, respectively. For wild-type and mutated RAF, SnRK2 plasmids, 3 μg per transfection were used. After transfection, protoplasts were incubated for 5 h under light in washing and incubation solution (0.5 M mannitol, 20 mM KCl, 4 mM MES, pH 5.7) with or without 5 μM ABA. The mutations were introduced into wild-type RAFs using the primers listed in Supplementary Data 6.

**Generation of anti-pS171-SnRK2.6 antibody**. The anti-pS171-SnRK2.6 antibody was generated by ABcloneal. Phosphopeptide C-KSSVLHpSQPK was synthesized and used as an antigen to immunize rabbit and generate the polyclonal anti-phosphorylation antibody. Phosphorylation-non-specific antibody was removed using peptide C-KSSVLHSQPK.

**Immunoblotting**. 30 mg seedling samples of wild-type, $OK^{100}$-oct, and $OK^{100}$-nonu were ground into fine powder in liquid nitrogen. Total proteins were extracted in 100 μL protein extraction buffer (100 mM HEPES, pH 7.5, 5 mM EDTA, 5 mM EGTA, 10 mM DTT, 10 mM $Na_3VO_4$, 10 mM NaF, 50 mM β-glycerophosphate, 1 mM PMSF, 5 μg/mL leupeptin, 5 μg/mL antipain, 5 μg/mL aprotinin, and 5% glycerol). Cell debris was removed by centrifugation at 12,000 g at 4 °C for 40 min and supernatant was collected. 20 μg protein were separated by 10% SDS/PAGE, and the proteins were transferred to PVDF membrane. Blots were probed with primary antibodies against SnRK2.2/2.3/2.6 (Agrisera) at a dilution of 1:5000, p-S175-SnRK2.6 at a dilution of 1:5000[33], and p-S171-SnRK2.6 (ABclonal) at a dilution of 1:5000. Anti-actin antibody (ABclonal) was used as the loading control at a dilution of 1:10,000. Secondary anti-rabbit antibodies at a dilution of 1:20,000 were used to detect antibodies in conjugation with secondary horseradish peroxidase and enhanced chemoluminescence reagent (ShengEr).

**Reporting summary**. Further information on research design is available in the Nature Research Reporting Summary linked to this article.

## Data availability

The RNA sequencing data were deposited to the GEO database with the dataset identifier GSE152691. Source data are provided with this paper.

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

## Acknowledgements

This work was supported by the Strategic Priority Research Program of the Chinese Academy of Sciences, Grant XDB27040106 (to P.W.), and National Natural Science Foundation of China, Grant 91017001 and 31771358 (to P.W.). We are grateful to Prof. Pedro Rodriguez of Universidad Politecnica de Valencia, Spain for kindly providing the *pyl112458* mutant seeds. We thank Life Science Editors for editorial assistance.

## Author contributions

P.W. designed the research. Z.L., Y.L., Y.W., X.L., L.M., Z.Z., C.M., and Y.Z. performed the experimental studies. Z.L., Y.W., X.L., S.X., P.L., and P.W. carried out the analysis. C.-P.S., H.S., J.-K.Z., and P.W. wrote the manuscript.

## Competing interests

The authors declare no competing interests.
