## [Peer Review File · Nature Communications]

REVIEWER COMMENTS

Reviewer #1 (Remarks to the Author):

Previous studies have shown that multiple mutants of several B2/B3 Raf-like kinases displayed ABA-responses (Katsuta et al., 2020; Lin et al., 2020; Takahashi et al., 2020), leading to the conclusion that the dissociation of SnRK2 kinases from PP2C phosphatases via RCAR receptors in an ABA-dependent manner is enough to activate the SnRK2s.

By contrast, in the present manuscript, the mutants of most B2/B3 Raf-like kinases (OK100-oct and OK100-nonu) showed ABA-insensitive phenotypes similar with the mutants of RCAR receptors and SnRK2 kinases (Figure 3, 4). On this basis the authors suggest that the existence of Raf-like kinases are essential to prime and activate SnRK2s in response to ABA (Figure 6).

It is difficult to properly assess the manuscript given that the mutant analyses were not comprehensive. The authors have previously shown that the mutant (OK-quatdec; Lin et al., 2020), which lacks B2/B3 Raf-like kinases (B3-RAF3,4,5 and B2-RAF7,8,9,10) in the mutant background of B4 Raf-like kinases, showed decreased but not completely impaired ABA responses. The mutations of B4 Raf-like kinases might have only minor effects on ABA responses. By contrast, in the present manuscript, the OK100-oct mutant, which lacks B3-RAF3,4,5 and B2-RAF7,8,9,10,11 (Figure 2A), showed nearly complete ABA-insensitivities (Figure 3, 4).

A question that remains is if the phenotypic differences can be explained by RAF11. RAF11 was shown to activate ABA responses in protoplasts (Figure 4E), whilst the result was unclear when one SnRK2 was co-transformed (Figure 4F). Ideally, the mutants of Raf-like kinase candidate should be chosen based on in vitro and in protoplasts experiments (Figure 1A, 4E, 4F), indicating that B3-RAF3,4,5,6 and B2-RAF7,10 might be important, in order to reveal which Raf-like kinases are involved in the activation of the SnRK2s. That said it is largely unclear which B2/B3 Raf-like kinases play predominant roles.

Furthermore, the data presented in Figure 5 is not convincing. Given that B2/B3 Raf-like kinases (OK100) were not activated by ABA (Figure 3E, 5A), a question is whether - and if so - how these kinases phosphorylate and activate the SnRK2s in response to ABA. The authors showed that the mutations in the activation loop of RAF10 did not impair its kinase activity to the SnRK2 (Figure 5D), implying that RAF10 does not require prior activations. That is, once PP2Cs are dissociated, RAF10 can phosphorylate the SnRK2s. However, this was not the case for RAF3 (Figure 5C, D). These two cases therefore did not explain the ABA-insensitive phenotypes of the mutants (OK100-oct and OK100-nonu; Figure 3, 4) or convincingly support the model (Figure 6).

Collectively, the conclusion is highly interesting and might allow one to extend or revise our previous hypotheses, however, the experiments did not seem to be carefully enough designed in order to support the conclusion. I feel that given the substantial support for the prevailing view (mentioned above), this study needs to provide further lines of evidence in support of their novel hypothesis.

Surprisingly, Katsuta et al., 2020 was not cited.

References:

Katsuta et al., Plant J. 2020 Jul;103(2):634-644. doi: 10.1111/tpj.14756; Lin et al., Nat Commun. 2020 Jan 30;11(1):613. doi: 10.1038/s41467-020-14477-9;

Takahashi et al., Nat Commun. 2020 Jan 2;11(1):12. doi: 10.1038/s41467-019-13875-y.

Reviewer #2 (Remarks to the Author):

The article by Lin et al. Initiation and amplification of SnRK2 activation in abscisic acid signaling aims to clarify how SnRK2 kinases are quickly activated during ABA signaling by the B2 and B3 RAFs. They show *in vivo* evidence of transphosphorylation of the SnRK2.6 which complements an already known *in vitro* characteristic. Authors further highlight that this transphosphorylation by RAFs is essential for the SnRK2 activation through creating and testing high-order Arabidopsis mutants lacking multiple B2 and B3 RAFs showing ABA hyposensitivity in seed germination and altered seedling growth, stomatal movement, and transcriptional gene regulation. They also show that basal level activation of B2 and B3 RAFs is a necessity and sufficient for SnRK2 activation in response to ABA. Overall, their work highlights an important *in vivo* contribution on the role of RAFs in SnRK2s activation.

Here I highlight some concerns towards improving this work for publication

Major comments

With the given experimental data, it is still unclear which of the RAFs in the B2 or B3 is critical in the ABA signaling either in the seed and the seedling or growing plant. This is a major concern that I feel needs to be addressed since the high order mutants do not clarify this. Besides the possibility of redundancy, we still observe that some RAFs are more effective than others and this is important to reveal in this study especially that the authors recommend this study output towards application in crop plants.

Other comments.

1. italicize terms '*in vivo*' and '*in vitro*'

Results

1. Line 132-134 these statements need further elaboration. In particular to what extent is Ser 171 and Ser175 important in SnRK2.6 phosphorylation. It is not clear how trans-phosphorylation by RAFs and SnRK2.6 autophosphorylation clearly require the two serines.
2. Line 147 "Preincubation with RAF3 significantly enhanced the kinase activity of recombinant SnRK2.6M94G" This is an interesting aspect however it still remains unclear how RAF3 enhance kinase activity of SnRK2.6. Does it phosphorylate other sites that promote downstream SnRK2 autophosphorylation. What is the real part of the mechanism that RAF is playing? This has been briefly described in the discussion but a link is then necessary for clarity.
3. Line 153- in the concluding statement "Thus, transphosphorylation by RAFs is required for the re-activation of dephosphorylated SnRK2.6" what is the evidence of trans-phosphorylation? It is important to show this or state it here to have a solid call that RAF is a vital factor.
4. Would it be correct to say B3 subgroups not critical for ABA signaling in seeds?
 - This study generalise the role of RAFs in plants not taking into consideration that some of the RAF could be tissue or rather organ specific. For example, the RAF B2 could be more seed specific than the B3 and OK130. I encourage the authors to look into this aspect and incorporate their thoughts and findings in this manuscript.
 - On the same note, how does the B2 RAFs mutants behave in response to stress in the seedlings or grown up plants? how is this comparable to ABA hyposensitivity seen on the seeds during germination.
 - Also since OK100-B3 had an arrested growth in soil than B2, does it mean that B3 is more important after seed germination than is B2- i.e marking the switch in their role in activation of the SnRKs. Can the authors comment on this?
 - Also is seed dormancy affected in the RAF B2 OK100, that is do they observe a rapid release of dormancy compared to wild type?
5. line 186; The OK100-nonu mutant produced very few seeds and had a lower seed germination rate- should it not be that reduction in ABA signaling enhance germination rate? May the authors expand on this. Essentially the RAFs show pleiotropic effect right?
 - Besides, the OK100-oct and OK100-nonu do show essential role of B2 and B3 subgroups of RAF in ABA signaling but it is equally important that the authors pinpoint which of these B2 and B3 is actually more important for development and also for seed dormancy and germination

6. The fact that >600 ABA inducible genes could still be significantly up regulated inform us that there is still an active ABA signaling pathway. Can the authors explain the detection of these large amount ABA-inducible or repressible genes in the mutant lines.

7. Since co-expression of RAF3, 5 and 11 only could rescue ABA-induced expression of RD29B-LUC, why did the authors not investigate the mutants of these RAFs separately in place of oct and nonu? The RAFs 3,5 and 11 could actually be more interesting and provide a more targeted analysis

- Secondly, in your transient activation assays, authors did not include RAF 11, which leaves questions on the specificity of RAF11 on the SnRK2. Besides, in these assays, did the authors attempted RAF3 alone for SnRK2.6 and and RAF 5 for SnRK2.2? Could these rescue the ABA-induced RD29B-LUC expression?

8. What is the kinase responsible for the basal phosphorylation in RAF2 and 3? This is worth mentioning or postulating with minimal evidence.

Discussion

1. Since OK100-B3 had an arrested growth in soil than B2, does it mean that B3 is more important after seed germination than is B2- i.e marking the switch in their role in activation of the SnRKs. Can the authors comment on this?

2. Line 310-312: Can the authors comment on this statement' "Our findings regarding B2 and B3 RAFs in stress signaling provide new targets for engineering crops resistant to harsh environmental conditions." In particular on the feasibility in the case that we see a lot of redundancy as explained by their results and use of mutants with multiple members of the RAF family in order to see a visible phenotype

3. Line 313: Although the authors excluded RAF1/CTR1 from the OK100 high order mutants, did they try to mutate just the KD of RAF1 and observe its phenotype. Even if it has been shown by Kieber et al 1993 that ctr1 mutant displays a severe phenotype, it is worth considering this RAF1 as a potential key target for linking ABA and ethylene signaling.

- Not able to recover the ABA-induced expression of RD29B-LUC in the protoplasts of OK100-oct cannot rule out the critical role of RAF1 in SnRK2 activation. This part of the work requires further elucidation to substantiate this claim including why subcellular locations could impact localization of RAF1 and its activity in the protoplasts.

4. This work is interesting however, besides in vivo attempts, authors should have attempted to search for key RAFs in the ABA signaling. Considering redundancy found in the RAFs, it is equally conceivable that some of the RAFs are more effective than others and narrowing this down could be more useful especially towards bioengineering crops.

Figures

Figure 1: Legend requires more elaboration for example, in panel A authors should indicate what SnRK2.6 KR.S171A stands for etc

- Can the authors explain the two bands shown on ABF2 phosphorylation especially in Lane 10

Figure 1D- de-SnRK manipulation is not clear.

Figure 2C: the grey boxes should be clearly explained in the legend, it may be highly confusing for the reader. THE same applies for figure 3

Figure 2D- can the writing be put outside of the picture, as it is it is not easy to see the figure well

Figure 3F, Could you explain the two bands detected in anti-pS171 and in anti-pS175

Reviewer #3 (Remarks to the Author):

The work reported in this manuscript is a breakthrough that involves the detailed biochemical analysis of the SnRK2.6 activation by RAF kinases as well as novel insights on the role of RAF kinases in ABA signaling. Once active SnRK2.6 is generated, which involves at least two steps: i) relief of PP2Cs by PYLs, ii) activation by RAFs, subsequent phosphorylation of PP2C-free SnRK2.6 can be performed by already active SnRK2.6. The above findings open new avenues of research

for ABA signaling and plant response to abiotic stress in higher plants. However, further clarification on the mechanism of SnRK2.2/2.3 activation is required; otherwise, some claims of the manuscript are overstated and the full story is not complete.

SnRK2.6 generated in *E. coli* is produced in an active state, which involves phosphorylation of the activation loop and capability to autophosphorylate and transphosphorylate different targets of SnRK2.6, such as ABF2, ABI5 or SLAC1. However, several studies have shown that once dephosphorylated by clade A PP2Cs, SnRK2.6 is unable to autophosphorylate, and therefore, transphosphorylation of SnRK2s by Raf-like protein kinases (B2, B3-type MAPKKs) is required for SnRK2 activation. An additional complication arises when SnRK2.2 and SnRK2.3 are considered, which are hardly active compared to SnRK2.6 when produced in *E. coli*. However, SnRK2s produced in *E. coli* achieve formation of partially active conformations in the absence of activation loop phosphorylation (Ng et al. 2011). These authors studied the structural basis for basal activity and autoactivation of SnRK2s, and concluded that the structure of SnRK2.6 is prone to autophosphorylation, whereas the structure of SnRK2.2/2.3 not so much. More precisely, SnRK2.6 has well-structured activation loop phosphate acceptor sites that are positioned next to the catalytic site, thus providing a basis for efficient autophosphorylation. Ser175 phosphorylation was required for SnRK2.6 kinase activity *in vitro* and SnRK2.6 autophosphorylation was 10-fold higher than that of SnRK2.2/2.3.

Activation loop autophosphorylation can occur intermolecularly (in trans) or intramolecularly (in cis). The study reported by Lin et al., represents a breakthrough to understand the initiation and amplification of SnRK2.6 activation in ABA signaling. However, the authors should complete some experiments with SnRK2.2 or SnRK2.3 to further elucidate the mechanism with these kinases. The work assumes a similar mechanism for the three ABA-activated SnRK2s based on experiments with SnRK2.6, but a previous work (Ng et al., 2011), as well as evidence reported by others (Cai et al., 2014), indicates that basal activity of SnRK2s is different and this can affect the amplification mechanism proposed by the authors for all ABA-activated SnRK2s. For instance, phosphorylation of SnRK2.6 may be initiated by RAFs and propagated in trans for SnRK2.2/2.3. This has important implications for ABA signaling. Even though the three kinases are involved in ABA signaling, it seems that SnRK2.6 has a major role in stomatal response whereas SnRK2.2/2.3 plays a major role in root. Cai et al. found that BIN2 can phosphorylate T181 of SnRK2.2 (T180 in SnRK2.3), but is not able to phosphorylate SnRK2.6.

Figure 1. The authors analyze the capability of the kinase domain (KD) of different RAFs to phosphorylate SnRK2.6 (the K to R mutation is a dead form to avoid autophosphorylation). Both B3 (RAF1 to 6) and B2 (RAF7, 10, 11 and 12) RAFs were able to phosphorylate SnRK2.6KR. It seems that B3 RAFs show certain specificity to phosphorylate S175 (particularly clear for RAF2, 5 and 6). B3-dependent signal is observed in the S171A mutant; therefore B3-type phosphorylation likely represents Ser175-P. On the other hand, B2 RAFs still phosphorylate the S175A mutant of SnRK2.6 (practically as wt SnRK2.6), which suggests specificity for Ser171. Phosphorylation of Ser171A by B2 RAFs is abolished whereas a clear phosphorylation signal is detected in the case of B3 RAFs. This finding probably will lead to further research in the group, but at least a reference in the model should be included. At this stage, authors do not know whether both residues are simultaneously or sequentially phosphorylated by RAFs. So, this should be properly reflected in the model, which only considers a simultaneous phosphorylation of SnRK2 and does not include the differential specificity for Ser171 and Ser175 by B2 and B3 RAFs.

What is the *in vitro* specificity of RAF-KD? Can the authors provide a negative control for RAF-KD (i.e. a protein that is not phosphorylated by RAF-KD)? I mean RAF1/CTR1, which is localized in ER, is not likely a kinase of SnRK2s; however it efficiently phosphorylates SnRK2.6 *in vitro* (although intriguingly phosphorylation of SnRK2.6 by RAF1/CTR1 is abolished either by Ser171Ala or Ser175Ala mutations, which does not happen with the rest of RAFs)

Fig 1b. The authors show that GST-SnRK2.6 is able to autophosphorylate and transphosphorylate His-SnRK2.6KR. This result, as well as previous reports from different groups, suggests that active SnRK2.6 is able to transphosphorylate inactive molecules of SnRK2.6. What about SnRK2.2 and SnRK2.3? Is RAF-activated SnRK2.6 able to phosphorylate SnRK2.2/2.3?

Fig 1c. The authors devise a clever assay to distinguish trans- and autophosphorylation of SnRK2.6. The SnRK2.6M94G mutant is the only kinase able to use the ATP analog as a thiophosphate donor (neither RAF3 KD nor wt SnRK2.6 can use it). The authors demonstrate that SnRK2.6M94G has a basal activity to autophosphorylate or transphosphorylate other molecules of

SnRK2.6M94G, and this capability is strongly enhanced when RAF3-KD is added in the preincubation mixture with normal ATP. It is possible that preincubation with RAF3-KD leads preferentially to phosphorylation of Ser175 in SnRK2.6M94G, which in turn leads to autophosphorylation or transphosphorylation of SnRK2.6M94G in Ser171. On the other hand, the authors use RAF10-KD to activate SnRK2.6M94G, which leads presumably to phosphorylation of Ser171. In this case, autophosphorylation or transphosphorylation of SnRK2.6M94G should occur in Ser175

I think the authors are in position to further unveil the mechanism of autophosphorylation or transphosphorylation of SnRK2.6M94G once it is activated by B2 or B3 RAF, i.e. whether it proceeds via phosphorylation of Ser171 or Ser175, or both. To this end, they should generate SnRK2.6M94G S171A and SnRK2.6M94G S175A as controls of the reaction. Adequate reactions could provide a major advance on the molecular details of the phosphorylation by the use of these substrates together with either RAF3-KD or RAF10-KD.

Lines 153-155. I think these lines need further clarification. After transphosphorylation by RAFs, SnRK2.6 is already phosphorylated probably both at Ser171 and 175. Therefore, I do not see how SnRK2.6 can quickly auto-phosphorylate. Instead, it seems more logical that active SnRK2.6 transphosphorylates (intermolecularly) more SnRK2.6 molecules that were not activated by RAFs yet. Current writing seems confusing

Figure 2. The authors investigate the contribution of B2 or B3 RAFs to activation of ABA signaling. It was previously established that B4 RAFs are UAKs of osmostress-activated subfamily I SnRK2s. B2s play apparently a predominant role over B3s because the OK100-B2 mutant shows more ABA insensitivity than the OK100-B3 mutant. However, it seems that OK100-B3 has a basal defect in germination that is enhanced in 1 μ M ABA (compared to wt, OK100-B3 seems to be 'ABA hypersensitive?'). Apparently this result is contradictory with results reported by Takashahi et al. (2020), who isolated the raf3 raf4 raf5 triple mutant as ABA-insensitive in a germination screening. Additionally, Takashahi et al. (2020) reported that amiRNA_m3k (an artificial microRNA line that impairs expression of 5 B3-RAF) shows ABA insensitive phenotype. The authors should discuss these different results with respect to ABA sensitivity of B3-raf mutants, particularly whether the raf2 allele is responsible of such variation.

Figure 2e is probably an in-gel kinase assay, but it is not indicated either in the text or figure legend. The authors state that ABA-induced phosphorylation is markedly reduced in OK100-B2 compared to wt; however this cannot be appreciated in the upper panel. I understand that quantification has been performed using the anti-SnRK2s panel, but in this case the immunoblot for analysis of Col-0 cannot be separated from the immunoblot performed for analysis of OK100-B2 and OK100-B3 (a right presentation is used in Figure 3f for the analysis of OK100-oct and OK100-nonu mutants compared to Col-0). Both Figure 2e and Figure 3f need a better description in the legend and description of the quantification.

Figure 3. The authors have generated the OK100-oct and OK100-nonu mutants, which contain null mutations in eight and nine RAF genes, respectively. These mutants show strong ABA insensitivity. I think the authors have omitted the introduction of the raf2 allele to avoid the 'ABA-hypersensitive' phenotype of OK100-B3. Please, discuss this point; otherwise it seems a paradox compared to OK100-B3 phenotype

I understand and appreciate the formidable task performed by the authors to generate the above genetic resources. I agree that particularly OK100-nonu mutant shows a strong ABA insensitive phenotype, which is one step below the snr2.2/2.3/2.6 triple or 112458 sextuple mutants (panel at 25 μ M ABA). Therefore, residual activation of SnRK2s likely occurs in OK100-nonu. This is likely observed upon overexposure of the film in figure 3e. Residual activation might be performed by the missing RAFs (RAF1, RAF2, RAF12), but the authors show in Figure 4E that these kinases cannot rescue the ABA-induced expression of RD29B-LUC in the OK100-oct. Therefore, alternative explanations are possible. According to Figure 3f, it is apparent a marked upregulation of SnRK2 levels in OK100-oct and OK100-nonu mutants (both minus or plus ABA), leading to SnRK2 levels that might escape from titration by PP2Cs. This point should be mentioned and discussed in the text.

If only B2/B3 RAFs were involved in phosphorylation of SnRK2.2/2.3/2.6, the OK100-nonu should show an identical ABA-insensitive phenotype to snrk2.2/2.3/2.6 triple mutant at high ABA concentration (which is not the case, see Figure 3D at 10-25 μ M ABA).

Fig. 4. Why have the authors used the OK100-oct instead of the OK100-nonu? I guess some growth phenotype of OK100-nonu might interfere with these analyses.

Fig. 5. ABA does not activate B2 and B3 RAFs, but osmotic stress induces the phosphorylation of the activation loop in RAF2 and RAF3. The authors generate a non-phosphorylatable mutation of RAF3, which is not able to activate SnRK2.6. In contrast, a similar mutation in RAF10 is able to activate SnRK2.6. An interesting aspect of this study is the comparison to PpARK, also known as PpCTR1 (Yushumura et al., 2015), where phosphorylation of Ser1029 is induced by ABA. Ser1029 is equivalent to one of the phosphorylated residues of the RAF3 activation loop (please, indicate in the text), but evidence for phosphorylation of the RAF10 activation loop is not provided in this study (Figure 5A). If such evidence was obtained in previous study by Lin et al, please, indicate proper reference.

Finally, I strongly suggest that the PpARK/PpCTR1 double nomenclature is used in discussion because it reflects convergent signaling of ET and ABA in *Physcomitrella*. Both Saruhashi et al. and Yushumura et al. published the same year (2015) their reports on PpARK and PpCTR1, respectively. These references should be included after lines 320-321

Discussion, line 290. Same comment as above. I do not see how SnRK2s can quickly auto-phosphorylate when they have been activated by RAFs. Instead, it seems more logical that active SnRK2.6 transphosphorylates (intermolecularly) more SnRK2.6 molecules that were not activated by RAFs yet. Additionally, the authors extend this mechanism to SnRK2.2/2.3, but as described above, experiments are lacking for these kinases.

Minor points.

The nomenclature used for 112458 changes along the manuscript. I suggest that at first citation is defined as pyr1 pyl1 pyl2 pyl4 pyl5 pyl8, and abbreviated as 112458 for the rest of the study. Vlad et al., 2009 (*Plant Cell*) should be cited in line 70-71. Indeed both Vlad et al., (2009) and Umezawa et al., (2009) were the first ones to report dephosphorylation of Ser175-P by PP2Cs. Moreover, in line 68, Ma et al., 2009 and Park et al., 2009 are not appropriate references for PP2Cs as negative regulators of SnRK2s. Please, replace by Vlad et al., 2009 and Umezawa et al., 2009

A mention to the work of Cai et al., 2014 (*PNAS*) should be included

Point to point response to reviewers' COMMENTS

Reviewer #1 (Remarks to the Author):

Previous studies have shown that multiple mutants of several B2/B3 Raf-like kinases displayed ABA-responses (Katsuta et al., 2020; Lin et al., 2020; Takahashi et al., 2020), leading to the conclusion that the dissociation of SnRK2 kinases from PP2C phosphatases via RCAR receptors in an ABA-dependent manner is enough to activate the SnRK2s. By contrast, in the present manuscript, the mutants of most B2/B3 Raf-like kinases (OK100-oct and OK100-nonu) showed ABA-insensitive phenotypes similar with the mutants of RCAR receptors and SnRK2 kinases (Figure 3, 4). On this basis the authors suggest that the existence of Raf-like kinases are essential to prime and activate SnRK2s in response to ABA (Figure 6).

It is difficult to properly assess the manuscript given that the mutant analyses were not comprehensive. The authors have previously shown that the mutant (OK-quatdec; Lin et al., 2020), which lacks B2/B3 Raf-like kinases (B3-RAF3,4,5 and B2-RAF7,8,9,10) in the mutant background of B4 Raf-like kinases, showed decreased but not completely impaired ABA responses. The mutations of B4 Raf-like kinases might have only minor effects on ABA responses. By contrast, in the present manuscript, the OK100-oct mutant, which lacks B3-RAF3,4,5 and B2-RAF7,8,9,10,11 (Figure 2A), showed nearly complete ABA-insensitivities (Figure 3, 4).

(Reviewer 1 Comment 1): *A question that remains is if the phenotypic differences can be explained by RAF11. RAF11 was shown to activate ABA responses in protoplasts (Figure 4E), whilst the result was unclear when one SnRK2 was co-transformed (Figure 4F). Ideally, the mutants of Raf-like kinase candidate should be chosen based on in vitro and in protoplasts experiments (Figure 1A, 4E, 4F), indicating that B3-RAF3,4,5,6 and B2-RAF7,10 might be important, in order to reveal which Raf-like kinases are involved in the activation of the SnRK2s. That said it is largely unclear which B2/B3 Raf-like kinases play predominant roles.*

Response: We agree with the reviewer that, ideally, the candidates for mutation should be chosen based on the *in vitro* or *in vivo* assays. Thus, in our previous study (Lin et al., 2020), we generated the *OK¹⁰⁰-quin* mutant with null mutations in five RAFs showing osmotic-stress-induced phosphorylation. As the *OK¹⁰⁰-quin* only showed a very weak ABA insensitivity, we then generated the *OK¹⁰⁰-B2* and *OK¹⁰⁰-B3* high-order mutants that knock-out Rafs in the B2 or B3 subgroup, respectively. However, the ABA insensitivity of *OK¹⁰⁰-B2* and *OK¹⁰⁰-B3* is still weaker than that of *pyl112458* and *snrk2-triple* mutants. We further spent one and a half years to generate the additional *OK¹⁰⁰-oct* and *OK¹⁰⁰-nonu* that are nearly completely ABA-insensitive and comparable to *pyl112458* and *snrk2-tiple*. We have no shortcut for generating mutations in certain genes by genome-editing. For each single high-order mutant, we need to screen millions of seeds and sequence hundreds or even thousands of seedlings in up to 8 generations to get the homozygous line. We cannot be sure which combination of mutations we get, and we also cannot predict the phenotype of these mutants before we finally perform the tests. We spent more than four years and more than 2,000,000 RMB (about \$300,000) generating the *OK¹³⁰-weak*, *OK¹³⁰-null*, *OK¹⁰⁰-quin*, *OK¹⁰⁰-B2*, *OK¹⁰⁰-B3*, *OK¹⁰⁰-oct*, *OK¹⁰⁰-nonu*, *OK-quatdec*, and *OK-quindec*, etc, as we believed these mutants would be valuable resources to us as well as the research community. Due to limited students, funding, lab resources, as well as the low efficiency of some guide RNAs, we could not get all the high-order mutants we wanted. We hope the reviewer can understand that and agree that we can only work with the high-order mutants we already have.

Following the reviewer's suggestion, we included *OK-quatdec* (B4-16,18,20,35,40, B3-RAF3,4,5, B2-RAF7,8,9,10) in the ABA sensitivity assay to further dissect the role of RAF11 in ABA signaling (updated Fig. 4). Compared to *OK¹⁰⁰-oct* (B3-RAF3,4,5,B2-RAF7,8,9,10,11), the *OK-quatdec* has wild type RAF11 and seven more mutations in B4 subgroup RAFs, which may not be involved in the activation of ABA-dependent SnRK2s (updated Fig. 4). *OK-quatdec* showed a similar ABA sensitivity to *OK¹⁰⁰-oct* in the context of germination and seedling growth. The *raf11* single mutant only showed a very weak ABA insensitivity (Lee et al., 2014), compared to *OK¹⁰⁰-B2*, *OK¹⁰⁰-B3* high-order mutants. These results indicate that RAF11 alone does not determine strong ABA insensitivity. The B2 and B3 Rafs have redundant functions in ABA signaling,

and the mutants carrying more knock-out mutations showed stronger ABA insensitivity than mutants carrying fewer mutations.

To further evaluate which B2/B3 Raf-like kinases have a predominate role in ABA-regulated germination and seedling development, we backcrossed *OK¹⁰⁰-nonu* to Col-0 and harvested the F2 seeds. We genotyped 103 F2 seedlings that could germinate on 1/2 MS medium, 1% sucrose, with 10 μ M ABA. As shown in the new **Table S1**, our result suggested that **RAF3, RAF4, RAF5, and RAF7/8/9 (closely linked)** might have predominant roles in germination and seedling establishment. We hope this new piece of data answers the reviewer's question. However, we would like to note that this result does not mean that these RAFs also have dominant roles in other ABA-regulated processes. **RAF3, RAF5, RAF7, and RAF11** (Fig. 5e, f) showed more robust activity in rescuing ABA-induced *RD29B-LUC* expression in *OK¹⁰⁰-oct* protoplasts. As shown in Supplementary Fig. 5 in our previous paper (Lin et al., 2020), different RAFs have different expression patterns, and they may participate in various biological processes. This might be why higher plants have multiple members in the B2 and B3 subgroup of RAFs (12 in Arabidopsis, and 9 in rice). We revised the result and discussion parts according to these new results.

(Reviewer 1 Comment 2): Furthermore, the data presented in Figure 5 is not convincing. Given that B2/B3 Raf-like kinases (*OK100*) were not activated by ABA (Figure 3E, 5A), a question is whether - and if so - how these kinases phosphorylate and activate the *SnRK2s* in response to ABA. The authors showed that the mutations in the activation loop of *RAF10* did not impair its kinase activity to the *SnRK2* (Figure 5D), implying that *RAF10* does not require prior activations. That is, once PP2Cs are dissociated, *RAF10* can phosphorylate the *SnRK2s*. However, this was not the case for *RAF3* (Figure 5C, D). These two cases therefore did not explain the ABA-insensitive phenotypes of the mutants (*OK100-oct* and *OK100-nonu*; Figure 3, 4) or convincingly support the model (Figure 6).

Response: There are two pieces of data strongly supporting the idea that B2 and B3 Rafs were not activated by exogenous ABA but kept a basal level of activity even without any stress: 1) The in-gel kinase assay result showed no clear induction of *OK¹⁰⁰* activity upon ABA treatment, but there is an apparent activity of *OK¹⁰⁰* in Col-0, *pyl112458*, *snrk2-triple*, even without any treatment (Fig. 4f, see the highlighted region in the figure below); Phosphoproteomics showed that several phosphosites in the activation loop of *RAF2/3* could be detected even without any treatment, but their phosphorylation is not upregulated by ABA (Fig. 6a). In Fig. 1 and the newly added Fig. 2, we showed that: 1) RAFs are required for the re-activation of dephosphorylated *SnRK2.2/2.3/2.6* (Fig 1b and 1f); 2) pre-activated *SnRK2.6* intermolecularly transphosphorylates *SnRK2.6^{KR}*, *SnRK2.2^{KR}*, and *SnRK2.3^{KR}* (Fig. 2a, Supplementary Fig. 2b); 3) *SnRK2* transphosphorylation can further activate *SnRK2.6^{MG}*, in the context of thiophosphorylation of *SnRK2.6^{MG}* and *ABF2* (Fig. 2d). We believe these lines of evidence already strongly support our model in Figure 6 (Fig. 7 in the revision).

In Fig. 5c and 5d, we showed that Ser to Ala mutations of these phosphosites only abolished the kinase activity of *RAF3*, but not that of *RAF10*. That might be because *RAF10* activation is not dependent on phosphorylation. As there are 12

members in the B2 and B3 subgroup of RAFs in Arabidopsis, it is reasonable that different RAFs may have different activation mechanisms. In animal cells, RAF kinases can be activated through phosphorylation, dimerization, or by the binding of small GTPase, scaffold protein, 14-3-3 protein, etc. (please see the reviews Lavoie & Therrien, 2015, Nat Rev Mol Cell Biol; Zoi et al., Nat Rev Cancer; Chong et al., 2003, Cellular Signaling). We added some discussion to elaborate on this notion in the revision.

(Reviewer 1 Comment 3): *Collectively, the conclusion is highly interesting and might allow one to extend or revise our previous hypotheses, however, the experiments did not seem to be carefully enough designed in order to support the conclusion. I feel that given the substantial support for the prevailing view (mentioned above), this study needs to provide further lines of evidence in support of their novel hypothesis.*

Response: In the revision, we added several lines of evidence:

- 1) The newly added Fig. 1b and Fig. 1f suggest that SnRK2s only have weak, or no phosphorylation activity and preincubation with RAF3-KD and RAF10-KD significantly enhances the kinase activity of SnRK2.2, 2.3, and 2.6.
- 2) The newly added Fig. 2d shows that pre-activated SnRK2.6 could transphosphorylate and activate SnRK2.6^{MG}, in the context of thiophosphorylation of SnRK2.6^{MG} and ABF2. These results further corroborate the RAF kinase-mediated initiation and the SnRK2 intermolecular transphosphorylation-mediated amplification of SnRK2 activation in ABA signaling.
- 3) We backcrossed *OK¹⁰⁰-nonu* to Col-0 wild type. By genotyping 103 F2 seedlings that could germinate on 10 μM ABA, we showed that RAF3-5 and RAF7-9 might have predominant roles in germination and seedling establishment.
- 4) We also performed additional experiments and added new data to address the more specific questions raised by other two reviewers. Please see the responses to the other reviewers' comments for details.
- 5) We also revised the discussion following the specific comments from the other two reviewers.

We believe our revised manuscript was improved by adding new lines of evidence and addressing the reviewer's constructive questions.

(Reviewer 1 Comment 4): *Surprisingly, Katsuta et al., 2020 was not cited.*

Response: We seem to have lost Katsuta et al., 2020 and some other references in transferring the manuscript between authors, which might be because of a technical issue with the Endnote software. We have double-checked the reference list, and Katsuta et al., 2020 and some more references are cited in the revision.

Reviewer #2 (Remarks to the Author):

The article by Lin et al. Initiation and amplification of SnRK2 activation in abscisic acid signaling aims to clarify how SnRK2 kinases are quickly activated during ABA signaling by the B2 and B3 RAFs. They show in vivo evidence of transphosphorylation of the SnRK2.6 which complements an already known in vitro characteristic. Authors further highlight that this transphosphorylation by RAFs is essential for the SnRK2 activation through creating and testing high-order Arabidopsis mutants lacking multiple B2 and B3 RAFs showing ABA hyposensitivity in seed germination and altered seedling growth, stomatal movement, and transcriptional gene regulation. They also show that basal level activation of B2 and B3 RAFs is a necessity and sufficient for SnRK2 activation in response to ABA. Overall, their work highlights an important in vivo contribution on the role of RAFs in SnRK2s activation. Here I highlight some concerns towards improving this work for publication.

Major comments:

(Reviewer 2 Comment 1): *With the given experimental data, it is still unclear which of the RAFs in the B2 or B3 is critical in the ABA signaling either in the seed and the seedling or growing plant. This is a major concern that I feel needs to be*

addressed since the high order mutants do not clarify this. Besides the possibility of redundancy, we still observe that some RAFs are more effective than others and this is important to reveal in this study especially that the authors recommend this study output towards application in crop plants.

Response: As mentioned in response to **Reviewer 1 comment 2**, to evaluate which B2/B3 Raf-like kinases have predominant roles in ABA signaling, we backcrossed *OK¹⁰⁰-nonu* to Col-0 wild-type and harvested the F2 seeds. We genotyped more than one hundred F2 seedlings that could germinate on 1/2 MS plate containing 10 μ M ABA and 1% sucrose. As shown in the newly added Table S1, these results suggested that RAF3-5, RAF7-9 have more predominant roles in ABA- regulated seed germination and seedling establishment. It is important to note that the contribution of these RAFs in germination is not identical to their ability to rescue *RD29B-LUC* expression in the transient assay in protoplasts. We have revised the results and discussion parts according to these new results.

Other comments.

(Reviewer 2 Comment 2): 1. italicize terms 'in vivo' and 'in vitro'

Response: Thank you for the suggestion, we have italicized 'in vivo' and 'in vitro' in the revised manuscript.

Results:

(Reviewer 2 Comment 3): 1. Line 132-134 these statements need further elaboration. In particular to what extent is Ser 171 and Ser175 important in SnRK2.6 phosphorylation. It is not clear how trans-phosphorylation by RAFs and SnRK2.6 autophosphorylation clearly require the two serines.

Response: We thank the reviewer for the constructive suggestion. We further elaborate on this point by the newly added Fig. 2. We showed that the two serine residues, Ser171 and Ser175, are major autophosphorylation sites of SnRK2.6 (Fig. 2c). In Dataset S1, we showed that both Ser171 and Ser175 in SnRK2.6 could be transphosphorylated (¹⁸O-phosphorylation sites) by RAFs or autophosphorylated by activated SnRK2.6 (phos-s site) by mass spectrometry assay. Making Ser171 and Ser175 non-phosphorylatable completely abolished the ability of SnRK2.6 to transphosphorylate SnRK2.6^{KR} (Fig. 2c). Supplementary Fig. 2a showed that Thr179 and Tyr182 also contribute to the SnRK2.6 autophosphorylation activity.

(Reviewer 2 Comment 4): 2. Line 147 "Preincubation with RAF3 significantly enhanced the kinase activity of recombinant SnRK2.6M94G" This is an interesting aspect however it still remains unclear how RAF3 enhance kinase activity of SnRK2.6. Does it phosphorylate other sites that promote downstream SnRK2 auto-phosphorylation. What is the real part of the mechanism that RAF is playing? This has been briefly described in the discussion but a link is then necessary for clarity.

Response: Please see our answer to **Reviewer 2 Comment 3**. In vitro kinase assay, and mass spectrometry assay showed that besides Ser171 and Ser175, six other serine/threonine residues, Ser29, Ser43, Ser71, Thr176, Thr179, Tyr182, could be phosphorylated by either SnRK2.6 or RAF3. As shown in Fig 1b and 1f, RAF3- and RAF10-mediated phosphorylation significantly enhanced SnRK2.6 kinase activity, and activated SnRK2.2 and SnRK2.3. In Fig. 2d, we further revealed that the pre-activated SnRK2.6 could activate SnRK2.6^{MG} activity. Thus, as shown in Fig. 7, RAFs' major role in ABA signaling is to phosphorylate SnRK2s, when they are released from inhibition of PP2Cs, to initiate SnRK2 activation.

(Reviewer 2 Comment 5):3. Line 153- in the concluding statement "Thus, transphosphorylation by RAFs is required for the re-activation of dephosphorylated SnRK2.6" what is the evidence of trans-phosphorylation? It is important to show this or state it here to have a solid call that RAF is a vital factor.

Response: In Fig. 1a, we showed that the B2 and B3 RAFs mainly (trans-) phosphorylate SnRK2.6^{KR} at Ser171 and Ser175. Mass spectrometry results showed that Ser171 and Ser175, as well as six other phosphosites, could be transphosphorylated by RAF3-KD (¹⁸O-phosphosites in Dataset S1). We also showed pre-dephosphorylated SnRK2.6^{M94G} could not re-activate itself after incubating with N6-Benzyl-ATPyS for 30 min (Fig. 1d). Adding RAF3-KD or RAF10-KD

transphosphorylates SnRK2.6 in the presence of ATP (Supported by Fig. 1a and Dataset S1). RAF-mediated transphosphorylation reactivates the pre-dephosphorylated inactive SnRK2.6^{M94G} in a time-dependent manner (Fig. 1d). As the result of reactivation, SnRK2.6^{M94G} is capable of thiophosphorylating itself and ABF2 using N6-Benzyl-ATPyS as the phosphate donor (Fig. 1e). We revised the description of the mass spectrometry assay to make this point clearer. Following the suggestion by Reviewer 3, we also added new data that RAF3-KD and RAF10-KD also could trans-phosphorylate SnRK2.2 and SnRK2.3, and that transphosphorylation activated SnRK2.2 and SnRK2.3 (Fig. 1f). These newly added data further support the notion that RAFs are essential for SnRK2 reactivation after they are released from inhibition by PP2Cs.

(Reviewer 2 Comment 6):4. Would it be correct to say B3 subgroups not critical for ABA signaling in seeds?

- This study generalise the role of RAFs in plants not taking into consideration that some of the RAF could be tissue or rather organ specific. For example, the RAF B2 could be more seed specific than the B3 and OK130. I encourage the authors to look into this aspect and incorporate their thoughts and findings in this manuscript.

- On the same note, how does the B2 RAFs mutants behave in response to stress in the seedlings or grown up plants? how is this comparable to ABA hyposensitivity seen on the seeds during germination.

- Also since OK100-B3 had an arrested growth in soil than B2, does it mean that B3 is more important after seed germination than is B2- i.e marking the switch in their role in activation of the SnRKs. Can the authors comment on this?

- Also is seed dormancy affected in the RAF B2 OK100, that is do they observe a rapid release of dormancy compared to wild type?

Response: We thank the reviewer for the constructive suggestions. Following the suggestions, we performed more experiments on germination (Fig. 3c, bottom panel, 3e, Supplementary Fig 4c, d), seed dormancy (newly added Supplementary Fig. 4e), and leaf yellowing (newly added Supplementary Fig. 7i, j). Interestingly, *OK100-B3* shows ABA insensitivity on 1/2 MS medium supplied with 1% sucrose, but not on the medium without sucrose (Fig. 3, Supplementary Fig. 4). The *OK100-B3* also has a predominant role in seed dormancy (Supplementary Fig. 4e). We also revised the results and discussion according to the new results.

(Reviewer 2 Comment 7):5. line 186; The *OK100-nonu* mutant produced very few seeds and had a lower seed germination rate- should it not be that reduction in ABA signaling enhance germination rate? May the authors expand on this.

Essentially the RAFs show pleiotropic effect right?

- Besides, the *OK100-oct* and *OK100-nonu* do show essential role of B2 and B3 subgroups of RAF in ABA signaling but it is equally important that the authors pinpoint which of these B2 and B3 is actually more important for development and also for seed dormancy and germination.

Response: The low germination phenotype was observed in other mutants that have reduced ABA signaling, like *snrk2-triple* (Takahashi et al., 2009), or *pyl-duodecuple* (Zhao et al., 2012). That might be because “*srk2d/e/l* (another allele of *snrk2-triple*) seeds were unable to complete seed development effectively” (page 1347, Takahashi et al., Plant Cell Physiol, 2009). Following the reviewer’s suggestions, we performed more experiments on germination, seed dormancy, and leaf yellowing. Interestingly, these results showed that *OK100-B3* shows ABA insensitivity on 1/2 MS medium supplied with 1% sucrose, but not on medium without sucrose. The *OK100-B3* also has a predominant role in seed dormancy. We also revised the results and discussion according to the new results.

(Reviewer 2 Comment 8): 6. The fact that >600 ABA inducible genes could still be significantly up regulated inform us that there is still an active ABA signaling pathway. Can the authors explain the detection of these large amount ABA-inducible or repressible genes in the mutant lines.

Response: Though 613 ABA-inducible genes are still up-regulated in *OK100-oct*, transcriptome results suggested the expression of these ABA-induced genes is strongly impaired in the *OK100-oct* mutants (see the heatmap in Fig. 5b). For

example, 331 out of the 613 ABA inducible genes in the *OK¹⁰⁰-oct* mutants showed at least 2-fold less induction than in Col-0. That information can be accessed in Dataset S2. In the *OK¹⁰⁰-oct* mutant, RAF6, perhaps also RAF2 and RAF12, still kept their functions. That might be the reason that there is still an active ABA signaling pathway. Supporting this notion, *OK¹⁰⁰-nonu* has a stronger ABA insensitivity than *OK¹⁰⁰-oct*.

(Reviewer 2 Comment 9): 7. Since co-expression of RAF3, 5 and 11 only could rescue ABA-induced expression of RD29B-LUC, why did the authors not investigate the mutants of these RAFs separately in place of oct and nonu? The RAFs 3,5 and 11 could actually be more interesting and provide a more targeted analysis.

Response: Please see our response to **Reviewer 1 comment 1**. In brief: 1) Due to limited funding and lab resources, we could not get all possible high-order mutants we wanted. Fig. 5f showed that besides RAF3, RAF5, and RAF11, other RAFs, like RAF4, RAF6, RAF7, and RAF9 might also contribute to the ABA-induced expression of RD29B-LUC, when co-transfected with different SnRK2s; 2) It is worth noting that the contribution of different RAFs to germination/seedling development is not identical to their contributions to regulating RD29B-LUC expression. Just like ABA receptor PYR1/PYLs/RCARs, different RAFs may have different roles in various biological processes. It would take at least one more year to get the suggested mutants to perform more targeted analysis, , and even the suggested mutants may still not be enough to clarify the issue since several other RAFs are likely also involved, so we feel this is beyond the scope of this manuscript.

(Reviewer 2 Comment 10) - Secondly, in your transient activation assays, authors did not include RAF 11, which leaves questions on the specificity of RAF11 on the SnRK2. Besides, in these assays, did the authors attempted RAF3 alone for SnRK2.6 and and RAF 5 for SnRK2.2? Could these rescue the ABA-induced RD29B-LUC expression?

Response: We are a little confused by the comment. In Fig. 5f, we co-transfected each and everyone of RAF1 to RAF12 with SnRK2.2, SnRK2.3, or SnRK2.6 in the mesophyll protoplasts of *OK¹⁰⁰-oct*. RAF3 alone with SnRK2.6 induced the RD29B-LUC expression at about 8-fold, while RAF5 with SnRK2.2 at about 13-fold relative to control. The two combinations mentioned by the reviewer rescued the ABA-induced RD29B-LUC expression in the mesophyll protoplasts of *OK¹⁰⁰-oct*. The raw data can be accessed via the source data.

(Reviewer 2 Comment 11) 8. What is the kinase responsible for the basal phosphorylation in RAF2 and 3? This is worth mentioning or postulating with minimal evidence.

Response: We agree with the reviewer that it is crucial to identify the upstream kinase or mechanism responsible for B2 and B3 Raf's basal phosphorylation. We tried several methods like yeast-two-hybrid, proximity labeling and examined the previous ABA-responsive phosphoproteomics. We have a long list of candidates, including RLK and CDPKs, and we are verifying their roles in RAF activation. Unfortunately, we currently have not identified a kinase that can activate RAFs in vitro and in vivo.

Discussion

(Reviewer 2 Comment 12) 1. Since *OK¹⁰⁰-B3* had an arrested growth in soil than B2, does it mean that B3 is more important after seed germination than is B2- i.e marking the switch in their role in activation of the SnRK2s. Can the authors comment on this?

Response: We thank the reviewer for the good suggestion. We added some discussion in the revision on the new data showing the diverse roles of B2 and B3 RAFs in different ABA-regulated biological processes.

(Reviewer 2 Comment 13) 2. Line 310-312: Can the authors comment on this statement' "Our findings regarding B2 and B3 RAFs in stress signaling provide new targets for engineering crops resistant to harsh environmental conditions." In

particular on the feasibility in the case that we see a lot of redundancy as explained by their results and use of mutants with multiple members of the RAF family in order to see a visible phenotype.

Response: We thank the reviewer for the comment and added some discussion in the revision “Our findings regarding B2 and B3 RAFs in stress signaling provide new targets (e.g., ectopic expression of a stress-inducible or constitutively-activated form of Rafs in guard cells) for engineering crops resistant to harsh environmental conditions”.

(Reviewer 2 Comment 14) 3. Line 313: Although the authors excluded RAF1/CTR1 from the OK100 high order mutants, did they try to mutate just the KD of RAF1 and observe its phenotype. Even if it has been shown by Kieber et al 1993 that *ctr1* mutant displays a severe phenotype, it is worth considering this RAF1 as a potential key target for linking ABA and ethylene signaling.

- Not able to recover the ABA-induced expression of RD29B-LUC in the protoplasts of OK100-*oct* cannot rule out the critical role of RAF1 in SnRK2 activation. This part of the work requires further elucidation to substantiate this claim including why subcellular locations could impact localization of RAF1 and its activity in the protoplasts.

Response: Following the reviewer’s suggestion, we co-transfected the kinase domain of RAF1/CTR1 (RAF1-KD) in the protoplasts of the OK100-*oct* mutant. As shown in Supplementary Fig. 8e, the co-transfection of RAF1/CTR1-KD enhances the RD29B-LUC expression more than full-length RAF1 (RAF1) does (about 2.5-fold, $p < 0.05$). However, it still cannot respond to either ABA or ACC. We revised the discussion according to the reviewer’s comment.

(Reviewer 2 Comment 15) 4. This work is interesting however, besides *in vivo* attempts, authors should have attempted to search for key RAFs in the ABA signaling. Considering redundancy found in the RAFs, it is equally conceivable that some of the RAFs are more effective than others and narrowing this down could be more useful especially towards bioengineering crops.

Response: We thank the reviewer for the constructive suggestion. As mentioned in response to **Reviewer 1 Comment 1**, to evaluate which B2/B3 Raf-like kinases have a predominate role in ABA signaling, we backcrossed OK¹⁰⁰-*nonu* to Col-0 and harvested the F2 seeds. We genotyped 103 individual F2 seedlings that could germinate on 1/2 MS medium containing 10 μ M ABA. As shown in newly added Table S1, our result suggested that RAF3-RAF5, and RAF7-9 (closely linked) have more dominant roles in ABA-regulated germination and seedling development. It is worth noting that the contribution of different RAFs in germination/seedling development is not identical to their contribution in regulating RD29B-LUC expression. For the purpose of bioengineering crops, it would require much work to try different RAF combinations with different promoters or mutations, which means a massive project in itself in the future.

Figures:

(Reviewer 2 Comment 16) Figure 1: Legend requires more elaboration for example, in panel A authors should indicate what SnRK2.6^{KR.S171A} stands for etc

Response: Thank you for the suggestion, we further indicated what SnRK2.6^{KR}, SnRK2.6^{KR-S171A}, SnRK2.6^{KR-S171A}, and SnRK2.6^{KR-AA} stand for in the Figure Legend.

(Reviewer 2 Comment 17) - Can the authors explain the two bands shown on ABF2 phosphorylation especially in Lane 10

Response: The two bands in Lane 10 should be some degradation fragments of GST-SnRK2.6^{M94G}. These degradation fragments also could be seen in the Coomassie blue staining result (Bottom panel, Lane 2, indicated by red circle in the figure below). After being transphosphorylated by RAF3-KD, the SnRK2.6^{M94G} has strong autophosphorylation activity and these degradation fragments might be thio-phosphorylated and detected by anti- γ -S immunoblot. We have clarified this in the revision.

(Reviewer 2 Comment 18) Figure 1D- de-SnRK manipulation is not clear.

Response: Thank you for the suggestion, we further indicated what de-SnRK2.6^{M94G} (pre-dephosphorylated SnRK2.6) stands for in the Figure Legend.

(Reviewer 2 Comment 19) Figure 2C: the grey boxes should be clearly explained in the legend, it may be highly confusing for the reader. THE same applies for figure 3

Response: Thank you for the suggestion, we added some explanation to the Figure Legend.

(Reviewer 2 Comment 20) Figure 2D- can the writing be put outside of the picture, as it is it is not easy to see the figure well

Response: Thank you for the suggestion, we have moved the writing to the left side of the picture.

(Reviewer 2 Comment 21) Figure 3F, Could you explain the two bands detected in anti-pS171 and in anti-pS175.

Response: The upper bands detected in both anti-pS171 and anti-pS175 results are non-specific bands, as they also present in the samples of *snrk2-triple* mutant. In addition, only the lower band matches the size of SnRK2s. Please see the revised Fig. 4f (original Figure 3F, also a long-exposed image below) for the details, the non-specific bands are indicated by arrows.

Reviewer #3 (Remarks to the Author):

The work reported in this manuscript is a breakthrough that involves the detailed biochemical analysis of the SnRK2.6 activation by RAF kinases as well as novel insights on the role of RAF kinases in ABA signaling. Once active SnRK2.6 is generated, which involves at least two steps: i) relief of PP2Cs by PYLs, ii) activation by RAFs, subsequent phosphorylation of PP2C-free SnRK2.6 can be performed by already active SnRK2.6. The above findings open new avenues of research for ABA signaling and plant response to abiotic stress in higher plants. However, further clarification on the mechanism of SnRK2.2/2.3 activation is required; otherwise, some claims of the manuscript are overstated, and the full story is not complete.

SnRK2.6 generated in *E. coli* is produced in an active state, which involves phosphorylation of the activation loop and capability to auto-phosphorylate and trans-phosphorylate different targets of *SnRK2.6*, such as *ABF2*, *ABI5* or *SLAC1*. However, several studies have shown that once dephosphorylated by clade A PP2Cs, *SnRK2.6* is unable to auto-phosphorylate, and therefore, transphosphorylation of *SnRK2s* by Raf-like protein kinases (B2, B3-type MAPKKs) is required for *SnRK2* activation. An additional complication arises when *SnRK2.2* and *SnRK2.3* are considered, which are hardly active compared to *SnRK2.6* when produced in *E. coli*. However, *SnRK2s* produced in *E. coli* achieve formation of partially active conformations in the absence of activation loop phosphorylation (Ng et al. 2011). These authors studied the structural basis for basal activity and autoactivation of *SnRK2s* and concluded that the structure of *SnRK2.6* is prone to autophosphorylation, whereas the structure of *SnRK2.2/2.3* not so much. More precisely, *SnRK2.6* has well-structured activation loop phosphate acceptor sites that are positioned next to the catalytic site, thus providing a basis for efficient autophosphorylation. Ser175 phosphorylation was required for *SnRK2.6* kinase activity *in vitro* and *SnRK2.6* autophosphorylation was 10-fold higher than that of *SnRK2.2/2.3*.

Activation loop autophosphorylation can occur intermolecularly (*in trans*) or intramolecularly (*in cis*). The study reported by Lin et al., represents a breakthrough to understand the initiation and amplification of *SnRK2.6* activation in ABA signaling.

(Reviewer 3 Comment 1) However, the authors should complete some experiments with *SnRK2.2* or *SnRK2.3* to further elucidate the mechanism with these kinases. The work assumes a similar mechanism for the three ABA-activated *SnRK2s* based on experiments with *SnRK2.6*, but a previous work (Ng et al., 2011), as well as evidence reported by others (Cai et al., 2014), indicates that basal activity of *SnRK2s* is different and this can affect the amplification mechanism proposed by the authors for all ABA-activated *SnRK2s*. For instance, phosphorylation of *SnRK2.6* may be initiated by RAFs and propagated *in trans* for *SnRK2.2/2.3*. This has important implications for ABA signaling. Even though the three kinases are involved in ABA signaling, it seems that *SnRK2.6* has a major role in stomatal response whereas *SnRK2.2/2.3* plays a major role in root. Cai et al. found that *BIN2* can phosphorylate T181 of *SnRK2.2* (T180 in *SnRK2.3*), but is not able to phosphorylate *SnRK2.6*.

Response: We thank the reviewer for the constructive suggestion. We performed more experiments to address this question. The newly added Fig.1f showed that *RAF3* and *RAF10* transphosphorylate and activate *SnRK2.2* and *SnRK2.3*, which further supported the essential role of RAFs in the initiation of *SnRK2* activation. The newly added Fig. 2d showed that *SnRK2.6* pre-activated by *RAF3/10-KD* transphosphorylates and activates *SnRK2.6^{M94G}* *in vitro*. However, as pointed out by the reviewer, the three ABA-activated *SnRK2s* may have different expression patterns in tissue distribution and we are not sure if this transphosphorylation between different *SnRK2s* may exist and have an important role in plants.

(Reviewer 3 Comment 2) Figure 1. The authors analyze the capability of the kinase domain (KD) of different RAFs to phosphorylate *SnRK2.6* (the K to R mutation is a dead form to avoid autophosphorylation). Both B3 (RAF1 to 6) and B2 (RAF7, 10, 11 and 12) RAFs were able to phosphorylate *SnRK2.6KR*. It seems that B3 RAFs show certain specificity to phosphorylate S175 (particularly clear for RAF2, 5 and 6). B3-dependent signal is observed in the S171A mutant; therefore B3-type phosphorylation likely represents Ser175-P. On the other hand, B2 RAFs still phosphorylate the S171A mutant of *SnRK2.6* (practically as wt *SnRK2.6*), which suggests specificity for Ser171. Phosphorylation of Ser171A by B2 RAFs is abolished whereas a clear phosphorylation signal is detected in the case of B3 RAFs. This finding probably will lead to further research in the group, but at least a reference in the model should be included. At this stage, authors do not know whether both residues are simultaneously or sequentially phosphorylated by RAFs. So, this should be properly reflected in the model, which only considers a simultaneous phosphorylation of *SnRK2* and does not include the differential specificity for Ser171 and Ser175 by B2 and B3 RAFs.

Response: We thank the reviewer for the constructive suggestion. As previously reported (Vlad et al., 2010; Soon et al., 2012), mutating either Ser171 or Ser175 resulted in a complete abolishment of *SnRK2.6* activity. At this moment, we have no method to verify whether RAFs simultaneously or sequentially phosphorylate Ser171 and Ser175. Following the

suggestion, we added S171 and S175 in the revised Fig. 7 (original Fig. 6) to indicate the specificity of B2 and B3 RAFs on these two residues.

(Reviewer 3 Comment 3) *What is the in vitro specificity of RAF-KD? Can the authors provide a negative control for RAF-KD (i.e. a protein that is not phosphorylated by RAF-KD)? I mean RAF1/CTR1, which is localized in ER, is not likely a kinase of SnRK2s; however, it efficiently phosphorylates SnRK2.6 in vitro (although intriguingly phosphorylation of SnRK2.6 by RAF1/CTR1 is abolished either by Ser171Ala or Ser175Ala mutations, which does not happen with the rest of RAFs)*

Response: In our recent paper (Lin et al., 2020), we already showed that RAF5, RAF6, RAF10, RAF24 and RAF40 cannot phosphorylate ABF2 (Lin et al., 2020, Fig. 5e). As the kinase domains of B2/B3 RAFs, including CTR1/RAF1, are highly conserved, it is not surprising that the RAF1/CTR1-KD can phosphorylate SnRK2.6 in vitro. Following this comment and **Reviewer 2 Comment 14**, we tested the effect of RAF1/CTR1-KD on ABA-induced *RD29B-LUC* expression in the transient activation assay in protoplasts. The result showed that co-transfection of neither the RAF1/CTR1-KD nor the full-length RAF1/CTR1 could rescue the ABA-induced *RD29B-LUC* expression in the protoplasts of *OK¹⁰⁰-oct*. This result suggested an unknown mechanism preventing RAF1/CTR1 from phosphorylating SnRK2s in plants, which would be interesting to study in the future.

(Reviewer 3 Comment 4) *Fig 1b. The authors show that GST-SnRK2.6 is able to auto-phosphorylate and trans-phosphorylate His-SnRK2.6^{KR}. This result, as well as previous reports from different groups, suggests that active SnRK2.6 is able to trans-phosphorylate inactive molecules of SnRK2.6. What about SnRK2.2 and SnRK2.3? Is RAF-activated SnRK2.6 able to phosphorylate SnRK2.2/2.3?*

Response: We thank the reviewer for the constructive suggestion. As shown in supplementary Fig. 2b, the pre-activated SnRK2.6 transphosphorylated SnRK2.2^{KR} and SnRK2.3^{KR}. However, as pointed out by the reviewer, the three SnRK2s may have different tissue-specific expression, and we are unsure if this transphosphorylation between SnRK2s exists or has an important role in plants.

(Reviewer 3 Comment 5) *Fig 1c. The authors devise a clever assay to distinguish trans- and autophosphorylation of SnRK2.6. The SnRK2.6M94G mutant is the only kinase able to use the ATP analog as a thiophosphate donor (neither RAF3 KD nor wt SnRK2.6 can use it). The authors demonstrate that SnRK2.6M94G has a basal activity to auto-phosphorylate or trans-phosphorylate other molecules of SnRK2.6M94G, and this capability is strongly enhanced when RAF3-KD is added in the preincubation mixture with normal ATP. It is possible that preincubation with RAF3-KD leads preferentially to phosphorylation of Ser175 in SnRK2.6M94G, which in turn leads to autophosphorylation or transphosphorylation of SnRK2.6M94G in Ser171. On the other hand, the authors use RAF10-KD to activate SnRK2.6M94G, which leads presumably to phosphorylation of Ser171. In this case, autophosphorylation or transphosphorylation of SnRK2.6M94G should occur in Ser175.*

I think the authors are in position to further unveil the mechanism of autophosphorylation or transphosphorylation of SnRK2.6M94G once it is activated by B2 or B3 RAF, i.e. whether it proceeds via phosphorylation of Ser171 or Ser175, or both. To this end, they should generate SnRK2.6M94G S171A and SnRK2.6M94G S175A as controls of the reaction. Adequate reactions could provide a major advance on the molecular details of the phosphorylation by the use of these substrates together with either RAF3-KD or RAF10-KD.

Response: We thank the reviewer for the constructive suggestion. In the newly added Fig. 2c, we generated SnRK2.6^{M94G-S171A} and SnRK2.6^{M94G-S175A} and found that mutation on Ser171 or Ser175 completely abolished the transphosphorylation activity of SnRK2.6^{M94G}, suggesting both residues are required for transphosphorylation activity. Besides, Thr179 and Tyr182 in SnRK2.6 might also contribute to the transphosphorylation activity of SnRK2.6^{M94G} (Supplementary Fig. 2a).

(Reviewer 3 Comment 6) Lines 153-155. I think these lines need further clarification. After transphosphorylation by RAFs, SnRK2.6 is already phosphorylated probably both at Ser171 and 175. Therefore, I do not see how SnRK2.6 can quickly auto-phosphorylate. Instead, it seems more logical that active SnRK2.6 trans-phosphorylates (intermolecularly) more SnRK2.6 molecules that were not activated by RAFs yet. Current writing seems confusing.

Response: We agree and thank the reviewer for the helpful suggestion. We revised the descriptions of SnRK2.6 transphosphorylation throughout the manuscript.

(Reviewer 3 Comment 7) Figure 2. The authors investigate the contribution of B2 or B3 RAFs to activation of ABA signaling. It was previously established that B4 RAFs are UAKs of osmostress-activated subfamily I SnRK2s. B2s play apparently a predominant role over B3s because the OK100-B2 mutant shows more ABA insensitivity than the OK100-B3 mutant. However, it seems that OK100-B3 has a basal defect in germination that is enhanced in 1 μ M ABA (compared to wt, OK100-B3 seems to be 'ABA hypersensitive'?). Apparently this result is contradictory with results reported by Takahashi et al. (2020), who isolated the *raf3 raf4 raf5* triple mutant as ABA-insensitive in a germination screening. Additionally, Takahashi et al. (2020) reported that *amiRNA_m3k* (an artificial microRNA line that impairs expression of 5 B3-RAFTs) shows ABA insensitive phenotype. The authors should discuss these different results with respect to ABA sensitivity of B3-*raf* mutants, particularly whether the *raf2* allele is responsible of such variation.

Response: We thank the reviewer for the constructive suggestion. Following this comment as well as the **Reviewer 2 Comment 6**, we performed more phenotypic assays and found that the difference in ABA insensitivity is dependent on the sucrose in the medium. We added the corresponding result and discussion in the revised manuscript.

(Reviewer 3 Comment 8) Figure 2e is probably an in-gel kinase assay, but it is not indicated either in the text or figure legend. The authors state that ABA-induced phosphorylation is markedly reduced in OK100-B2 compared to wt; however, this cannot be appreciated in the upper panel. I understand that quantification has been performed using the anti-SnRK2s panel, but in this case the immunoblot for analysis of Col-0 cannot be separated from the immunoblot performed for analysis of OK100-B2 and OK100-B3 (a right presentation is used in Figure 3f for the analysis of OK100-oct and OK100-nonu mutants compared to Col-0). Both Figure 2e and Figure 3f need a better description in the legend and description of the quantification.

Response: We apologize for not clarifying that Fig. 2e is an immunoblot result using anti-pS171, anti-pS175 phospho-specific antibodies in the Figure Legend, though we mentioned it in line 175 in the manuscript (line 171 in the original submission). For the immunoblot, 20 μ g of total proteins was loaded in each lane, which could be verified by the anti-actin immunoblot (bottom panel). The anti-SnRK2 immunoblot showed that ABA treatment quickly triggered SnRK2 degradation in Col-0. This rapid degradation also could be found in some previous reports (Fig. 2e and 3f in Takahashi et al., 2020, see the image below). Interestingly, the ABA-triggered SnRK2 degradation is partially or entirely abolished in *OK100-B2*, *OK100-B3*, or *OK100-oct* mutants (revised Fig. 3f and 4g) or by mutation of S171A (Fig. 2e and 3f in Takahashi et al., 2020, see the image below). That could be an unknown desensitization mechanism of ABA signaling, which depends on RAF-mediated SnRK2 transphosphorylation and is worth exploring in the future. We added a description in the revision.

Fig. 2e and 3f from Takahashi et al. (2020)

As the original Fig. 2e included *OK¹⁰⁰-nonu* (see below), to avoid misleading and make the figure easy to follow, we only showed the results of Col-0 wild type, *OK¹⁰⁰-B2*, and *OK¹⁰⁰-B3* mutants in the original manuscript. Following this suggestion, we repeated the experiment and replaced the image with a new one in the revised Fig. 3f.

(Reviewer 3 Comment 9) Figure 3. The authors have generated the *OK100-oct* and *OK100-nonu* mutants, which contain null mutations in eight and nine RAF genes, respectively. These mutants show strong ABA insensitivity. I think the authors have omitted the introduction of the *raf2* allele to avoid the ‘ABA-hypersensitive’ phenotype of *OK100-B3*. Please, discuss this point; otherwise it seems a paradox compared to *OK100-B3* phenotype

Response: We thank the reviewer for the thoughtful suggestion. As mentioned in Line 162-263, only *RAF1/CTR1* is “omitted” in the design of the CRISPR/CAS9 knock-out system. A guide RNA CATCGGAGGTGGTAGCCAG was used to knock-out *RAF2/EDR1* in the generation of *OK¹⁰⁰-B3* or *OK¹⁰⁰-oct/OK¹⁰⁰-nonu* (See the Dataset S5 for details). The *OK¹⁰⁰-B3* mutant, which contains a null mutation in *RAF2*, showed an ABA-insensitive phenotype, not the ABA-hypersensitive phenotype as *raf2/edr1* in germination. That might be because *RAF3-5* have a predominant role in ABA-mediated germination and seedling establishment (Table S1), which is sufficient to cover the relatively weak ABA-hypersensitivity of *raf2/edr1*. *OK¹⁰⁰-oct/OK¹⁰⁰-nonu* are the best mutants we have that contain the maximum number of null mutations in B2 and B3 RAF, not including *RAF2* mutation by chance. It is reasonable that *OK¹⁰⁰-oct/OK¹⁰⁰-nonu* mutants containing null alleles of *raf3-5* and *raf7-9* showed an extreme insensitivity to ABA. Following this suggestion, we added some discussion in the revision.

(Reviewer 3 Comment 10) I understand and appreciate the formidable task performed by the authors to generate the above genetic resources. I agree that particularly *OK100-nonu* mutant shows a strong ABA insensitive phenotype, which is one step below the *snrk2.2/2.3/2.6* triple or 112458 sextuple mutants (panel at 25 μ M ABA). Therefore, residual activation of SnRK2s likely occurs in *OK100-nonu*. This is likely observed upon overexposure of the film in figure 3e. Residual activation might be performed by the missing RAFs (*RAF1*, *RAF2*, *RAF12*), but the authors show in Figure 4E that these kinases cannot rescue the ABA-induced expression of *RD29B-LUC* in the *OK100-oct*. Therefore, alternative explanations are possible. According to Figure 3f, it is apparent a marked upregulation of SnRK2 levels in *OK100-oct* and *OK100-nonu* mutants (both minus or plus ABA), leading to SnRK2 levels that might escape from titration by PP2Cs. This point should be mentioned and discussed in the text.

Response: We thank the reviewer for the constructive suggestion. As mentioned in response to **Reviewer 3 Comment 8**, the up-regulated SnRK2 level in *OK¹⁰⁰-oct* and *OK¹⁰⁰-nonu* mutants could be a result of bypassing ABA-induced SnRK2 degradation. Although *OK¹⁰⁰-oct* and *OK¹⁰⁰-nonu* have higher abundance of SnRK2 proteins, the ABA-induced SnRK2 activation, as well as the Ser171 and Ser175 phosphorylation are completely abolished, as shown in Fig. 4e and 4f.

(Reviewer 3 Comment 10) If only B2/B3 RAFs were involved in phosphorylation of SnRK2.2/2.3/2.6, the *OK100-nonu* should show an identical ABA-insensitive phenotype to *snrk2.2/2.3/2.6* triple mutant at high ABA concentration (which is not the case, see Figure 3D at 10-25 μ M ABA).

Response: We thank the reviewer for the comment. We cannot exclude the possibility that other protein kinases like BIN2 (Cai et al., 2014) and BAK1 (Shang et al., 2016) are also involved in SnRK2 activation in ABA signaling. BIN2 is known to

phosphorylate the conserved serine residue corresponding to Ser179 in SnRK2.6 (Cai et al., 2014). We added some discussion in the revision and cited these papers.

(Reviewer 3 Comment 11) Fig. 4. Why have the authors used the OK100-oct instead of the OK100-nonu? I guess some growth phenotype of OK100-nonu might interfere with these analyses.

Response: As mentioned in the manuscript, the OK¹⁰⁰-nonu only produced very few seeds, and we could not get enough seeds when we performed the RNA sequencing assay. However, we repeated the qRT-PCR experiment, and the results showed that OK¹⁰⁰-nonu has a similar (RAB18 and COR15A) or even more dramatic reduction (KIN1 and RD29B) in ABA-induced gene expression compared to OK¹⁰⁰-oct (revised Fig. 5c).

(Reviewer 3 Comment 12) Fig. 5. ABA does not activate B2 and B3 RAFs, but osmotic stress induces the phosphorylation of the activation loop in RAF2 and RAF3. The authors generate a non-phosphorylatable mutation of RAF3, which is not able to activate SnRK2.6. In contrast, a similar mutation in RAF10 is able to activate SnRK2.6. An interesting aspect of this study is the comparison to PpARK, also known as PpCTR1 (Yushumura et al., 2015), where phosphorylation of Ser1029 is induced by ABA. Ser1029 is equivalent to one of the phosphorylated residues of the RAF3 activation loop (please, indicate in the text), but evidence for phosphorylation of the RAF10 activation loop is not provided in this study (Figure 5A). If such evidence was obtained in previous study by Lin et al, please, indicate proper reference.

Response: We thank the reviewer for the suggestion. In the revision, we further clarified that the relative intensity of RAF2/RAF3 phosphopeptide upon ABA treatment is obtained from our previous study (Wang et al., Mol Cell, 2018), and the relative intensity of RA2/RAF3 phosphopeptide upon mannitol treatment is obtained from our previous study (Lin et al., Nat Commun, 2020). The re-analysis result of these phosphoproteomics data is provided in Dataset S4.

(Reviewer 3 Comment 13) Finally, I strongly suggest that the PpARK/PpCTR1 double nomenclature is used in discussion because it reflects convergent signaling of ET and ABA in Physcomitrella. Both Saruhashi et al. and Yushumura et al. published the same year (2015) their reports on PpARK and PpCTR1, respectively. These references should be included after lines 320-321.

Response: We thank the reviewer for the suggestion. We cited the references and revised the discussion.

(Reviewer 3 Comment 14) Discussion, line 290. Same comment as above. I do not see how SnRK2s can quickly auto-phosphorylate when they have been activated by RAFs. Instead, it seems more logical that active SnRK2.6 transphosphorylates (intermolecularly) more SnRK2.6 molecules that were not activated by RAFs yet. Additionally, the authors extend this mechanism to SnRK2.2/2.3, but as described above, experiments are lacking for these kinases.

Response: We thank the reviewer for the suggestion. We revised the discussion and added results from SnRK2.2 and SnRK2.3 as the revised Fig. 1f and Supplementary Fig. 2a.

Minor points.

(Reviewer 3 Comment 15) The nomenclature used for 112458 changes along the manuscript. I suggest that at first citation is defined as pyr1 pyl1 pyl2 pyl4 pyl5 pyl8, and abbreviated as 112458 for the rest of the study.

Response: Thank you for the comment, we have made changes accordingly.

(Reviewer 3 Comment 16) Vlad et al., 2009 (Plant Cell) should be cited in line 70-71. Indeed both Vlad et al., (2009) and Umezawa et al., (2009) were the first ones to report dephosphorylation of Ser175-P by PP2Cs

Response: Thank you for the comment, we have added the references in the revision.

(Reviewer 3 Comment 17) Moreover, in line 68, Ma et al., 2009 and Park et al., 2009 are not appropriate references for PP2Cs as negative regulators of SnRK2s. Please, replace by Vlad et al., 2009 and Umezawa et al., 2009.

Response: Thank you for the comment, we have corrected the citations in the revision.

(Reviewer 3 Comment 18) A mention to the work of Cai et al., 2014 (PNAS) should be included.

Response: Thank you for the comment, we have included the reference.

REVIEWER COMMENTS

Reviewer #1 (Remarks to the Author):

I feel that the authors did an admirable job of answering or rebutting the comments from my first review. Whilst I would have liked all the mutants that they mentioned trying to get I do accept that at such higher order this is a large ask and probably can be reserved for a separate manuscript. This has become a very nice story and the extra work added is truly convincing

Reviewer #2 (Remarks to the Author):

he revised manuscript by Lin et al clearly demonstrates an important mechanism of Initiation and amplification of SnRK2 activation in abscisic acid signaling. In the revised version the author have carefully addressed the issues that I had raised before.

A minor concern on Fig 4 panel g: Although the authors have indicated that the arrow indicate non-specific bands especially for anti-pS171, this unspecific target seems to be differentially phosphorylated and thus I would like to suggest that the authors determine what this band is? I suggest cutting out the band from the blot and perform trypsin digestion and Mass spectrometry. This should not consume much time and effort.

Reviewer #3 (Remarks to the Author):

Most of the major questions have been properly answered and the new experiments have improved markedly the manuscript. Great work.

Some questions remaining:

1) Lines 191-192:

OK100-B3 showed weak ABA insensitivity in germination and seedling development on half MS medium without sugar

No ABA insensitivity is observed in the absence of sucrose. Please correct this phrase. Given that Takahashi et al., (2020) isolated an ABA-insensitive *raf3,4,5* triple mutant in a germination screening, it is worth discussing whether sucrose was present in that screening compared to this work

2) Lines 204-207. ABA-induced phosphorylation of conserved serine residues 205 corresponding to Ser171 and Ser175 in SnRK2.6 was markedly reduced in the OK100-B2, and slightly reduced in the OK100-B3 mutants, but did not change in OK130-null (Fig. 3f).

It is not convincing the slightly reduced phosphorylation in the OK100-B3 mutant, unless you refer to the ratio of phosphorylated protein versus total SnRK2 protein. If so, please indicate it in the text. Moreover, the effect might be observed in pS175 but perhaps not in pS171.

3) Line 365: Is *raf2* resistant to high sugar or is a mistake? If mistake, then *raf1/ctr1/sis1* should be written to refer to the same mutant

4) Line 372: only 3791112 shows arrested growth...in response to ABA (I guess)
Indeed, this pentuple mutant behaves as wt regarding ABA sensitivity. I do not think this pentuple mutant reflects functional diversity, just low expression of the affected receptors. In any case, please, complete the phrase.

5) Lines 374-376: Please, cite the specific role of PYR1 in pathogen response. Garcia-Andrade, J., Gonzalez, B., Gonzalez-Guzman, M., Rodriguez, P.L. and Vera, P. (2020). The Role of ABA in Plant Immunity is Mediated through the PYR1 Receptor. *Int. J. Mol. Sci.* 21, E5852

Point to point response to reviewers' COMMENTS

Reviewer #2 (Remarks to the Author):

he revised manuscript by Lin et al clearly demonstrates an important mechanism of Initiation and amplification of SnRK2 activation in abscisic acid signaling. In the revised version the author have carefully addressed the issues that I had raised before.

(Reviewer 2 Comment 1) *A minor concern on Fig 4 panel g: Although the authors have indicated that the arrow indicate non-specific bands especially for anti-pS171, this unspecific target seems to be differentially phosphorylated and thus I would like to suggest that the authors determin what this bands is? I suggest cutting out the band from the blot and perform trypsin digestion and Mass spectrometry. This should not consume much time and effort.*

Response: Following the suggestion, we cut the blot and performed the Mass spectrometry (MS) assay. We failed to identify any phosphoproteins or protein kinases from the cut blot, which might be because of the interference of a high amount of primary and secondary antibodies, and milk proteins used for blocking. Alternatively, we performed an IP-MS using untreated wild-type seedlings and the Anti-pS171 antibody. Three phosphoproteins were identified by IP-MS, includes a protein kinase about 38.2 kDa, AT3G46760.1. Two phosphorylation sites in AT3G46760.1, T215 and T219, were detected by MS. As the similar molecular weight and phosphorylation state, the AT3G46760 might be the non-specific phosphoprotein recognized by the Anti-pS171 antibody. These data were added as supplementary Fig. S5c and Supplementary Dataset 2.

Reviewer #3 (Remarks to the Author):

Most of the major questions have been properly answered and the new experiments have improved markedly the manuscript. Great work.

Some questions remaining:

1) Lines 191-192: OK100-B3 showed weak ABA insensitivity in germination and seedling development on half MS medium without sugar.

No ABA insensitivity is observed in the absence of sucrose. Please correct this phrase.

Given that Takahashi et al., (2020) isolated an ABA-insensitive raf3,4,5 triple mutant in a germination screening, it is worth discussing whether sucrose was present in that screening compared to this work

Response: We revised the phrase and added a sentence in the discussion that *miRNA-M3K* was screened with the presence of sucrose. We also cited the corresponding reference Hauser et al., 2013.

2) Lines 204-207. ABA-induced phosphorylation of conserved serine residues

205 corresponding to Ser171 and Ser175 in SnRK2.6 was markedly reduced in the OK100-B2, and slightly reduced in the OK100-B3 mutants, but did not change in OK130-null (Fig. 3f).

It is not convincing the slightly reduced phosphorylation in the OK100-B3 mutant, unless you refer to the ratio of phosphorylated protein versus total SnRK2 protein. If so, please indicate it in the text. Moreover, the effect might be observed in pS175 but perhaps not in pS171.

Response: We revised the sentence to "ABA-induced phosphorylation of conserved serine residues corresponding to Ser171 and Ser175 in SnRK2.6 was markedly reduced in the *OK¹⁰⁰-B2*, and relatively not affected in the *OK¹⁰⁰-B3* and *OK¹³⁰-null* mutants".

3) Line 365: Is raf2 resistant to high sugar or is a mistake? If mistake, then raf1/ctr1/sis1 should be written to refer to the same mutant.

Response: We apologize for the typo. It is corrected in the revision.

4)Line 372: only 3791112 shows arrested growth...in response to ABA (I guess)

Indeed, this pentuple mutant behaves as wt regarding ABA sensitivity. I do not think this pentuple mutant reflects functional diversity, just low expression of the affected receptors. In any case, please, complete the phrase.

Response: We thank the reviewer for the suggestion, we rephrased the sentence to make it more apparent in the revision.

5)Lines 374-376: Please, cite the specific role of PYR1 in pathogen response. Garcia-Andrade,J., Gonzalez,B., Gonzalez-Guzman,M., Rodriguez,P.L. and Vera,P. (2020). The Role of ABA in Plant Immunity is Mediated through the PYR1 Receptor. Int. J. Mol. Sci. 21, E5852

Response: We thank the reviewer for the suggestion, we cited the reference in the revision.